

# 1 Analysis of the phase space of the downburst that occurred on 25 2 June 2021 in Sânnicolau Mare (Romania)

Andi Xhelaj [1,], Massimiliano Burlando [1]
*[1] Department of Civil, Chemical and Environmental Engineering*
*Polytechnic School, University of Genoa, Via Montallegro 1, 16145 Genoa, Italy*
*Correspondence to:* Andi Xhelaj (andi.xhelaj@edu.unige.it)
**Abstract**. Downbursts winds, characterized by strong, localized downdrafts and subsequent horizontal straight-line
winds, presents significant risk to civil structures. The transient nature and limited spatial extent present measurements
challenges, necessitating analytical models for accurate understanding and predicting their action on structures. This study
analyzes the Sânnicolau Mare downburst event in Romania, from June 25, 2021, using a bi-dimensional analytical model
coupled with the Teaching Learning Optimization Algorithm (TLBO). The intent is to understand the distinct solutions
generated by the optimization algorithm and assess their physical validity. Supporting this examination is a damage survey
and wind speed data recorded during the downburst event. Employed techniques include agglomerative hierarchical
clustering with the K-means algorithm (AHK-MC) and principal component analysis (PCA) to categorize and interpret
the solutions. Three main clusters emerge, each displaying different storm characteristics. Comparing the simulated
maximum velocity with hail damage trajectories indicates that the optimal solution offers the best overlap, affirming its
effectiveness in reconstructing downburst wind fields. However, these findings are specific to the Sânnicolau Mare event,
underlining the need for a similar examination of multiple downburst events for broader validity.
KEYWORDS: Downburst analytical model, Metaheuristic optimization algorithm, Multivariate data analysis, Downburst
kinematic and geometric parameters, Damage survey.

## 21 1 Introduction

The wind climatology of Europe and several mid-latitude countries are primarily dominated by the presence of extra-
tropical cyclones and thunderstorms. The understanding of the formation and evolution of extra-tropical cyclones dates
back to the 1920s (Bjerknes and Solberg, 1922). The atmospheric boundary layer (ABL) winds generated during such
systems are well recognized, and their influence on structures has been extensively studied and coded starting from the
1960s (Davenport, 1961). These established models continue to be employed in contemporary engineering practice
(Solari, 2019). Thunderstorm winds known as "downburst" consists of a strong and localized downdraft of air generated
within a convective cell. These downdrafts after reaching the ground begins to spread horizontally, resulting in the
formation of the downburst gust front, also known as the downburst outflow. The presence of strong turbulent wind within
the downburst outflow poses significant risk to civil structures. Downburst may be generated by isolated thunderstorms,
with length scales less than few kilometers. Additionally, they can be originated from more complex convective systems
such as squall lines and bow echoes, in this case the spatial length scale which can potentially be affected by downbursts
or downburst clusters is in the order of hundreds of kilometers (Fujita, 1978, Hjelmfelt, 2007). The size of the downburst
outflow area of strong winds exhibits variability, leading to the classification of this phenomenon as either a microburst



or macrobust. A microburst is characterized by a strong outflow size that is less than 4 km, whereas a macroburst
corresponds to an outflow size of intense wind greater than 4 km (Fujita, 1985). For the past four decades, the study of
intense downburst wind and their impact on the built environment has constituted a prevailing subject of research in the
field of Wind Engineering (Letchford, 2002). Since downburst event have high frequency of occurrence, they can be
considered as one of the most severe meteorological phenomena. Thunderstorm, occurring at the mesoscale, exhibit
nonstationary behaviour. Their origin is due to an instable convection condition in the atmosphere and the resulting
horizontal wind profiles are significantly different from those usually observed in the ABL. From a statistical point of
view, wind velocities, characterized by a mean return period greater than 10 or 20 years, are often due to these phenomena
(Solari, 2014). The lack of a unified model for downburst outflows and their actions on structures, similar to Davenport's
(1961) model for extra-tropical cyclones, is primarily due to significant uncertainties arising by the inherent complexity
of downburst winds. Indeed, the transient nature and limited spatial extent of downbursts presents challenges in their
measurements and restrict the availability of an adequate number of test cases. In 2020, Xhelaj et. al. presented an
analytical model thought to simulate the bi-dimensional structure of downbursts. The model depends on 11 parameters
that are estimated using a global metaheuristic optimization algorithm described in Xhelaj et. al. (2022). The integration
between the analytical model and the optimization algorithm, as well as the estimation of the kinematic parameters of the
downburst outflow, is based on the Teaching Learning Based Optimization (TLBO) algorithm. The TLBO algorithm
operates with a population of solutions and emulates a teaching and learning activity through iterative process to attain
the best solution within the population (Rao et al., 2011). Due to the stochastic nature of the TLBO algorithm when
coupled with the analytical model, the procedure can produce different optimum (or best) solutions each time the
algorithm is executed. This variability arises from the initial random population of solutions generated at the beginning
of the algorithm and the intermediate transformations of the set of solutions carried out by the algorithm in order to
converge towards the best solution. This study aims to examine the characteristics of the optimal solutions obtained
through multiple runs of the optimization procedure. It seeks to investigate the variability of the best solutions when
applying the optimization algorithm to reconstruct the wind field during an intense downburst event. The main objective
is to assess the extent to which the solutions differ from each other and from the solution with the lowest objective function
value. Additionally, the study explores whether these alternative solutions can be considered physically valid, particularly
when additional data describing the downburst event is incorporated. The selected downburst event occurred in western
Timis region of Romania on 25 June 2021 and was produced during the passage over the town of Sânnicolau Mare of an
intense mesoscale convective system of bow echo type. This event was recorded by a bi-axial anemometer and
temperature sensor, both placed on a telecommunication tower 50 m above the ground level. The telecommunication
tower lies approximately 1 km south of Sânnicolau Mare. The downburst that occurred in Sânnicolau Mare was of
significant magnitude, resulting in extensive hail damage of the facades of numerous buildings within the city. Subsequent
to the occurrence of this intense event, a comprehensive damage survey was undertaken through a collaborative
partnership between University of Genoa (Italy) and the University of Bucharest (Romania). The survey (Calotescu et.,
al., 2022 and Calotescu et., al., 2023 (submitted)) pinpoints the GPS position of the buildings within the city that were
predominantly impacted by the downburst. Moreover, a comprehensive map illustrating the hail damage of the building
facades was generated. The map provides important information regarding the wind velocity experienced at urban scale,
which has been used to validate the reconstruction/simulation of the downburst by the optimization procedure.
The analysis of the different optimal solutions (i.e., the data set) generated by the optimization algorithm was conducted
through multivariate data analysis (MDA). This involved the joint application of cluster analysis and principal component



analysis to effectively examine and interpret the dataset. Cluster analysis (CA) is a data mining technique that groups similar solutions together, aiming to identify patterns in the data. It is commonly used in fields like meteorology and climatology to identify clusters of weather phenomena or geographical regions with similar weather patterns (Burlando et al., 2008; Burlando et al., 2009). Principal component analysis (PCA) is a mathematical technique used to decrease the dimensionality of a dataset while minimizing the loss of information within the data. This analysis is commonly used in meteorology and climatology to decrease the number of variables required for representing weather pattern or climate trends and to identify regions with similar weather patterns (Amato et. al., (2020); Jiang et. al., (2020)). Principal component analysis is utilized in this context to enhance the interpretation of the different optimal solutions.

The present work is structured in 7 Sections. Following the introduction, Section 2 provides a description of the monitoring system that acquired the full-scale measurement employed in this research. Section 3 provides a brief meteorological description of the downburst event in Sânnicolau Mare (Romania). Section 4 describes the data set employed for performing cluster analysis and principal component analysis. Section 5 describes the implementation of these analyses. Section 6 presents an in-depth account of the main results derived from the CA and PCA. In conclusion, Section 7 offers a summary of the principal findings derived from this research.

## 2 Monitoring system and data acquisition

The complete set of measurements employed in this research were obtained through a monitoring system installed in Romania. Relevant information of this monitoring network can be accessed in the publications by Calotescu et al., (2021), Calotescu and Repetto, (2022) and Calotescu et. al., (2023) (submitted). The monitoring network received funding from the THUNDERR Project (Solari et al., 2020), which was conducted by the "Giovanni Solari – Wind Engineering and Structural Dynamics" Research (GS-Windyn) Group at the Department of Civil, Chemical, and Environmental Engineering (DICCA) of the University of Genoa. GS-Windyn, with a keen interest in monitoring poles and towers exposed to thunderstorm actions worldwide, secured funding for the acquisition of a full-scale structural monitoring network. This monitoring system was deployed on top of a 50 m lattice tower. The primary focus of this project revolves around three key objectives: first, the detection of thunderstorms; second, the analysis of wind parameters associated with these phenomena; and third, the experimental assessment of the structural response of telecommunication lattice towers to the forces generated by both synoptic and thunderstorm winds. Thunderstorms are local phenomena that occur in conditions of atmospheric instability, being characterized by the existence of vertical air currents that lead to the development of cumulonimbus clouds, the production of electric discharges, rain, and hail as well as strong downdrafts inducing damaging winds in proximity to the Earth's surface. The vertical profile of horizontal wind velocity in downburst winds showcases distinct characteristics when compared to the traditional velocity profile observed within the boundary layer. Notably, downburst winds exhibit a nose-like shape profile, with a pronounced maximum intensity near the ground. This specific profile presents a considerable risk, particularly for structures of low to medium height. The monitoring tower, named TM_424, is property of the SC TELEKOM ROMANIA SRL and is located in the western part of Romania, Timis county, at approximately 1 km south of Sânnicolau Mare (Figure 1). The site is an open field, the terrain is flat with low grass vegetation.




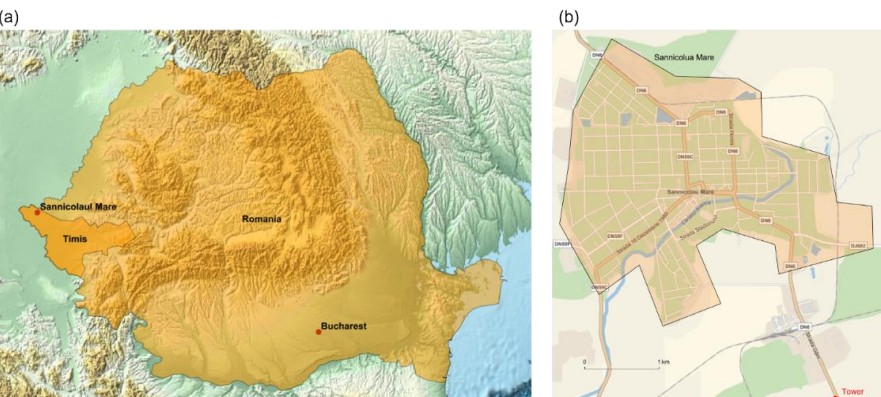


**Figure 1. (a) Location of the telecommunication tower TM_424, situated 1 km south of Sânnicolau Mare in Timis County, Romania. (b) Expanded view of the Sânnicolau Mare town with the telecommunication tower TM_424 represented by the red dot on the map. Maps generated using Mathematica (Wolfram Research, Inc., Version 13.3, 2023, https://www.wolfram.com/mathematica).**


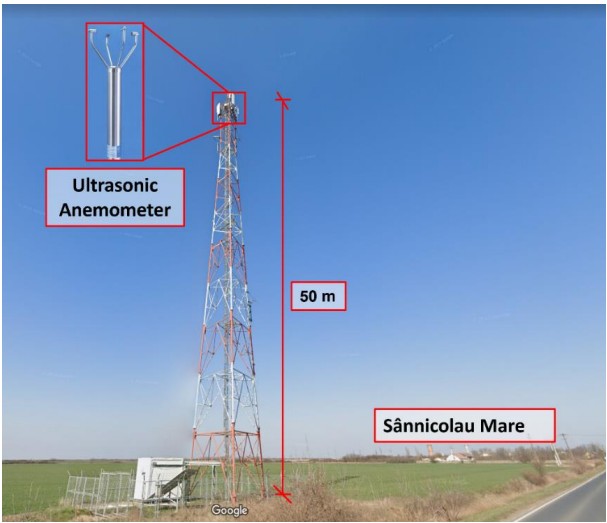

**Figure 2. TM_424 Telecommunication tower and sensors position at the top of the tower. On the horizon, approximately 1 km from the tower lies the small city of Sânnicolau Mare. Image courtesy of © Google Street View, 2022 (https://www.google.com/maps).**


Figure 2 shows the dimension of the tower. Among the various networks for the monitoring systems, the tower is
equipped with a GILL WindObserver 70 ultrasonic anemometer at the top (Figure 2). The anemometer has a data



acquisition rate of 4 Hz, can measure the wind speed up to 70 m/s. In addition to the anemometer sensor, the tower is
equipped with a temperature sensor installed near the location of the anemometer. The sensor was encased by a protective
case. The working temperature range for this sensor is between -55 and 70 °C.

**3 The Sânnicolau Mare (Romania) downburst event of 25 June 2021**
In this section, a brief overview of the meteorological aspects pertaining to the downburst event in Sânnicolau Mare on
June 2021 is provided. In the late afternoon of 25 June 2021, a severe downburst event affected the extreme western
region of Romania. The downburst event took place in the Timis county (Figure 1a) between 18:00 and 19:00 UTC and
struck the little town of Sânnicolau Mare (Figure 1b). At 17:30 UTC, a strong mesoscale convective system moving
toward the east was approaching the town of Sânnicolau Mare. Figure 3a, acquired from Eumetsat, captures an image of
a deep convective cell at 18:30 UTC. This weather phenomenon exhibits cloud tops ascending over 12 km above mean
sea level, signifying the mature stage of the convection cycle. This mature storm cell was observed to have directly
impacted the town under study. Figure 3b presents composite radar reflectivity data, indicating that this meteorological
phenomenon can be classified as a mesoscale convective system known as bow echo. Radar reflectivity values at or above
60 dBZ, as seen in this event, are typically indicative of severe weather conditions. Such conditions are often associated
with the production of hailstones, with an average diameter of approximately 2.5 cm.

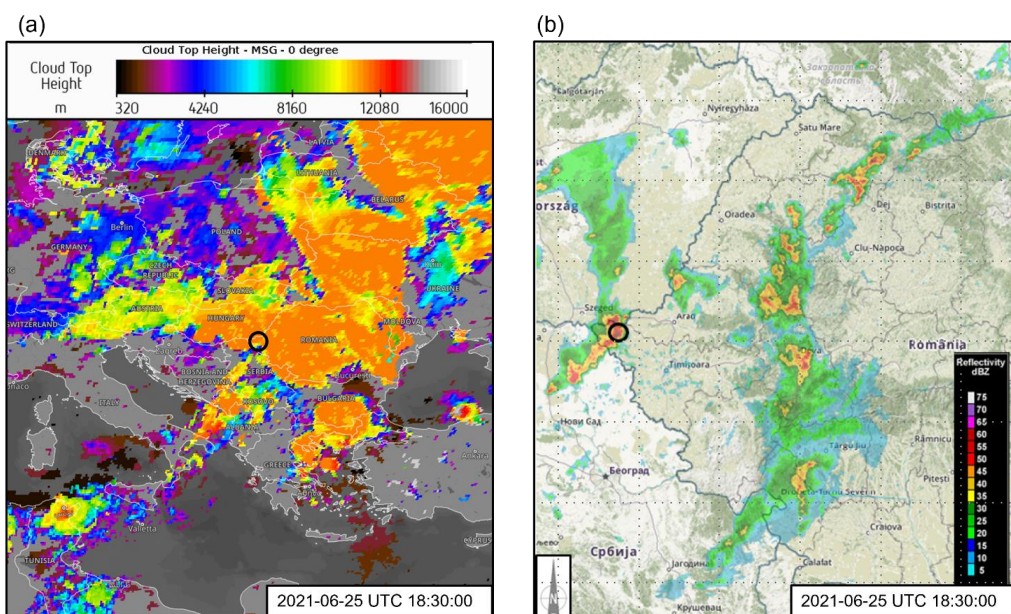

**Figure 3. (a) Distribution of cloud top heights derived from Meteosat Second Generation (MSG) valid for 25 June 2021 at 18:30**
**UTC. Data and map obtained from ©EUMETSAT 2022 (https://view.eumetsat.int). (b) Composite radar reflectivity (dBZ) for**
**June 25, 2021, at 18:30 UTC. The geographical location of Sânnicolau Mare and the apex of the bow echo are indicated by the**
**black circle. Data and map obtained by ©2018 Administratia Nationala de Meteorologie (https://www.meteoromania.ro).**
The existence of a robust convective motion, indicative of the typical kinematic structure of a bow echo, is distinctly
portrayed through the distribution of intensive lightning activity, as displayed in Figure 4a. As the figure illustrates, an
approximate total of 10455 lightning strikes were recorded by the Blitzortung.org network across Eastern Europe between



16:30 to 18:30 UTC. A significant concentration of these strikes correlates with the bow echo structure   near the western
Timis County in Romania. Bow echoes are a prevalent form of severe convective organization. These mesoscale
convective systems can generate straight-lines surface winds that lead to extensive damage associated with downbursts.
On occasion, they may also give rise to tornadoes.

(a)

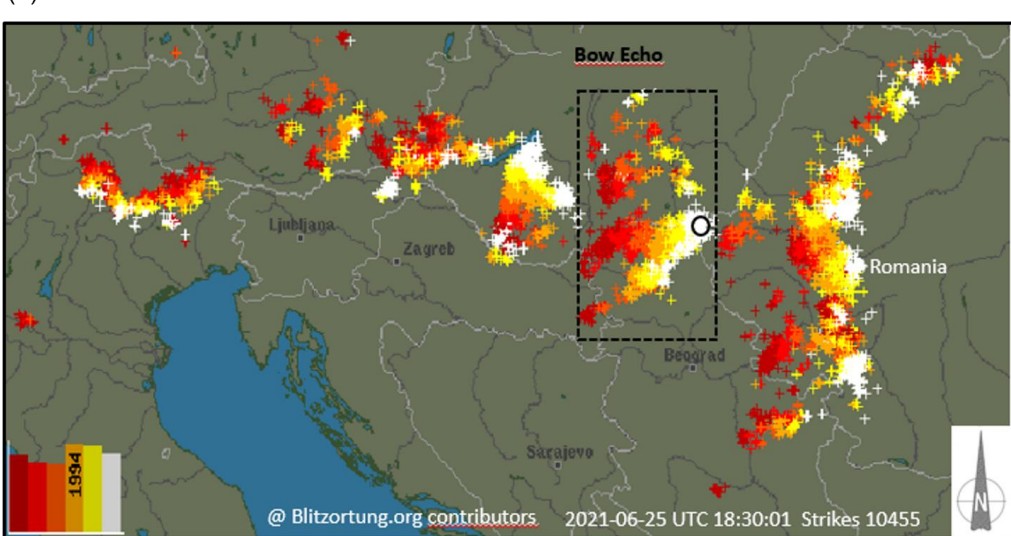

(b)

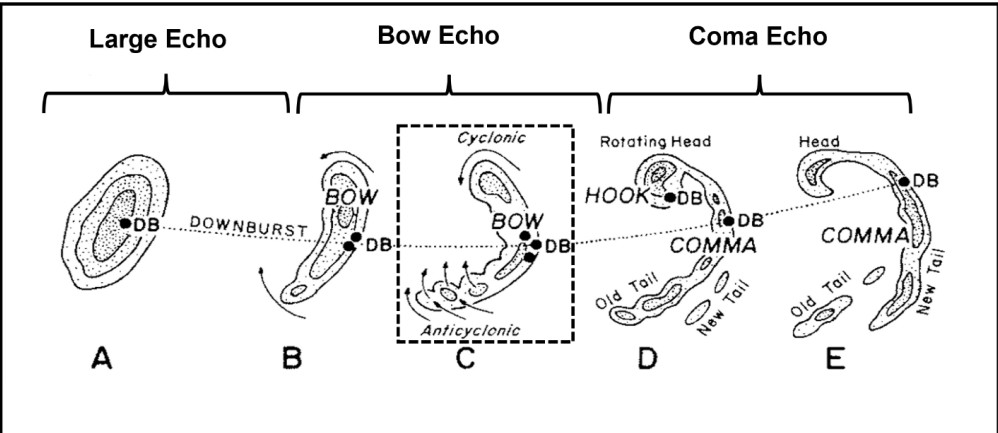

**Figure 4. (a) Lightning strikes recorded between 16:30 to 18:30 UTC on June 25, 2021, sourced from the Blitzortung.org**
**network archive for lightning and thunderstorms (www.blitzortung.org). The black circle marks the geographic location of**
**Sânnicolau Mare, situated near the apex of the observed bow echo. (b) Typical radar echo morphology commonly observed in**
**bow echoes, characterized by the generation of strong downbursts at the bow apex, denoted as DB. Adapted from Fujita (1978).**
Figure 4b illustrates the characteristic kinematic structure of a bow echo as outlined by Fujita (1978). Typically, the
system originates as a singular, prominent convective cell, either isolated or embedded within a broader squall line system
(Phase A). As the surface winds strengthen, the parent cell undergoes transformation, evolving into a line segment of cells



with a bow-shaped configuration (Phase B). During the maximum intensity, the bow's center might develop a spearhead
echo (Phase C), characterized by the occurrence of the most severe downburst winds at the apex of the spearhead. During
the decay phase, the wind system frequently evolves into a comma-shaped echo (Phase E) (Weisman, 2001). The
comparisons between Figures 3b, 4a, and 4b elucidate that the bow echo positioned above Sânnicolau Mare at 18:30 UTC
is in its most intense stage (Phase C), as evidenced by the formation of the characteristic spearhead echo shape. The
intense downburst event generated at the apex of the bow echo was recorded by the anemometer and temperature sensor
situated 50 meters above the ground on the TM_424 tower. The time histories of the moving average wind speed and
direction (averaged over 30 seconds) (Solari et al., 2015; Burlando et al., 2017) for the recorded one-hour duration of the
downburst event are given in Figure 5a and Figure 5b, respectively. At approximately 18:30 UTC the anemometer
recorded an instantaneous maximum velocity (sampled at 4 Hz) of $\hat{V} = 40.8$ m/s while the maximum moving average
wind velocity was $V_{\max}, = 35.8$ m/s. This notable high velocity clearly evidences of the occurrence of an intense
downburst. The time interval spanning from 18:20 to 18:45 UTC represents the primary indicator of the downburst's
occurrence in the proximity of the telecommunication tower. This period is characterized by a sudden surge in wind speed,
commonly referred intensification stage followed by a subsequent decrease in velocity after 18:30 UTC. Throughout the
initial phase of intensification, the wind direction exhibited a clockwise rotation, ranging from 235° and extending to
approximately 360°. Additionally,  Figure 5a also includes 1-hour time series of the recorded temperature data. The
temperature sensor is positioned at the same location of the anemometer. Before the passage of the downburst, the
environmental temperature was on average 27 °C, while at approximately 18:20 UTC the temperature dropped very
sharply reaching the minimum value of 14.5 °C at approximately 18:30 UTC. After the sharp drop the temperature started
to rise and eventually returned to its pre-storm level  (not shown).



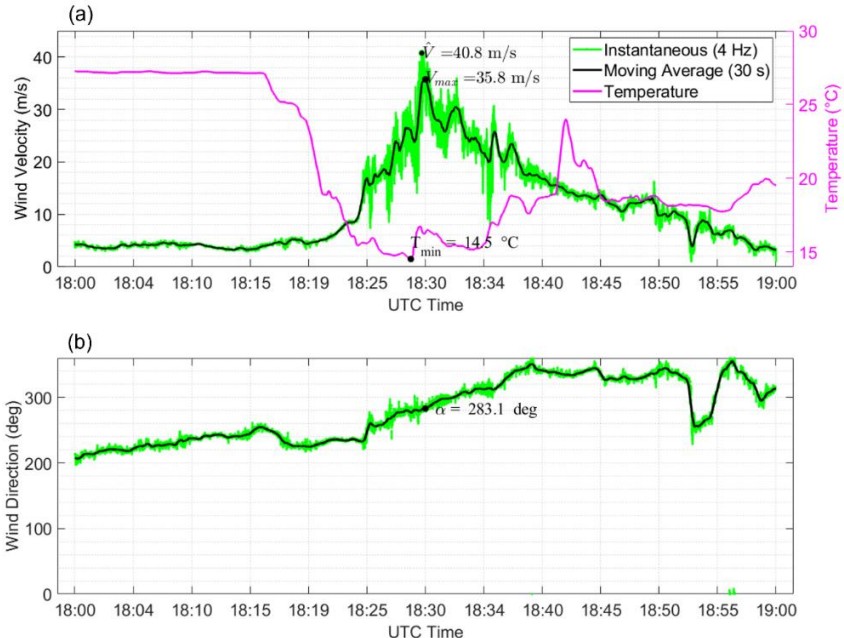

**Figure 5. Telecommunication tower monitoring network measurements from 18:00 to 19:00 UTC on June 25, 2021: (a) Time history of the instantaneous wind speed (green), moving average mean wind speed (black) and temperature record (magenta); (b) Instantaneous (green) and moving average mean wind direction (black).**


The downburst in Sânnicolau Mare caused widespread hail damage to the facades of numerous buildings. A collaborative
damage survey was conducted by the University of Genoa (Italy) and the University of Bucharest (Romania) (Calotescu
et al., 2022; Calotescu et al., 2023, submitted). The survey identified the affected buildings and produced a comprehensive
map illustrating the hail damage. Figure 6 shows a schematic representation of the distribution of hail damage per area
(600 x 600 m$^2$) and the position of the buildings that suffers hail damage in the town of Sânnicolau Mare.



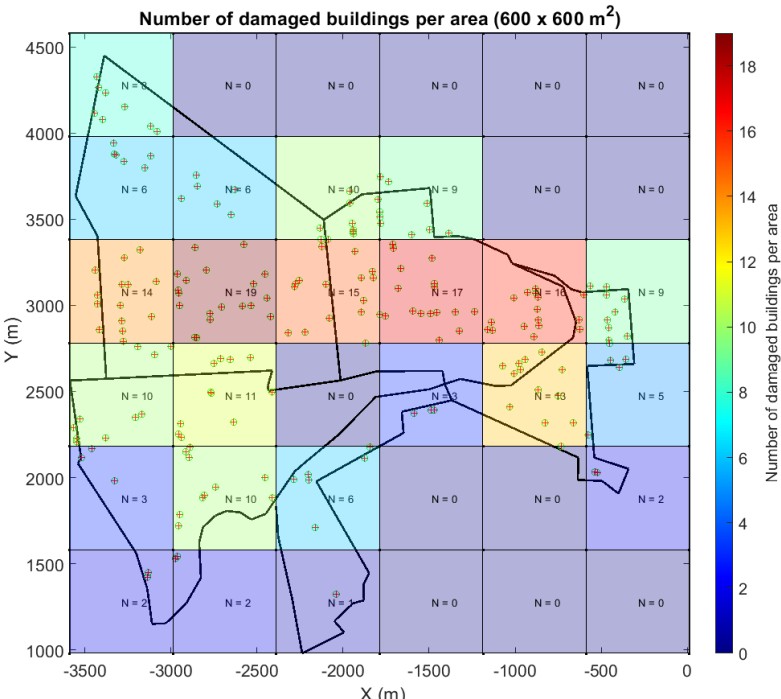


**Figure 6. Spatial distribution of damaged buildings and locations of hail-damaged structures within 600 x 600 m² area in the town of Sânnicolau Mare during the downburst event on June 25, 2021. The city boundaries of Sânnicolau Mare are delimited by the black line.**

**4 Downburst reconstruction**

This section focuses on the modeling, optimization, and reconstruction of the Sânnicolau Mare downburst event. Section 4.1 delves into the modeling and optimization approach used for downburst reconstruction. Section 4.2 introduces metaheuristic optimization and its application in the reconstruction of the specific downburst event under study. Finally, Section 4.3 outlines the multivariate data analysis employed to examine the solutions generated by the optimization algorithm.

**4.1 Modeling and optimization approach for downburst reconstruction**

In this study, the authors utilize the computational model developed in a previous work by Xhelaj et al. (2020) for the reconstruction and simulation of the Sânnicolau Mare downburst event discussed in Section 3. The Xhelaj et al. (2020) model can simulate the spatiotemporal evolution of the bi-dimensional moving average (30 second window) wind speed and direction experienced during a typical downburst event at a specified height z above ground level (AGL). The Xhelaj et al., (2020) model is able to reconstruct/simulate the space-time evolution of the bi-dimensional moving average wind speed and direction produced during a generic downburst event at a height z above the ground level (AGL). The wind system simulated by the model represents the outflow structure of a translating downburst embedded in a synoptic scale wind, which is considered as constant across the simulation domain. The model comprises 11 variables that describe the kinematic structure of the downburst wind. Table 1 presents a short description of the 11 variables upon which the model



relies. As a result, the model allows for the reconstruction of the time-evolving moving average wind speed and direction
generated by the simulated downburst at every point within the simulation domain. By employing anemometric wind
speed and direction data collected during the Sânnicolau Mare downburst event, an optimization procedure can be
formulated to minimize the relative error (objective function $F$), which quantifies the discrepancy between the observed
time series of the moving average wind speed and direction and the corresponding simulations generated by the model.
Since the Sânnicolau Mare downburst event was recorded by an anemometer positioned at a height of 50 meters AGL,
the analytical model will reconstruct the wind speed and direction at the corresponding height.
**Table 1. Variables of the Xhelaj et., al. (2020) analytical model.**

| | | |
|---|---|---|
| 1 | X-component touchdown location (at $t = 0$) $(m)$ | $X_{C0}$ |
| 2 | Y-component touchdown location (at $t = 0$) $(m)$ | $Y_{C0}$ |
| 3 | Downdraft radius $(m)$ | $R$ |
| 4 | Normalized radial distance from the center of the downburst where $V_{r,max}$ occurs $(-)$ | $\rho = \dfrac{R_{max}}{R}$ |
| 5 | Maximum radial velocity $(m/s)$ | $V_{r,max}$ |
| 6 | Duration of the intensification period $(min)$ | $T_{max}$ |
| 7 | Total duration of the downburst event $(min)$ | $T_{end}$ |
| 8 | Storm translational velocity $(m/s)$ | $V_t$ |
| 9 | Storm translational direction (deg) | $\alpha_t$ |
| 10 | ABL wind speed below the cloud base $(m/s)$ | $V_b$ |
| 11 | ABL wind direction below the cloud base $(deg)$ | $\alpha_b$ |


The reconstruction procedure gives rise to a mathematical optimization problem characterized by being single-objective,
nonlinear, and bound constrained, as discussed in Xhelaj et al. (2022). To tackle this optimization problem, the analytical
model is integrated with a global metaheuristic optimization algorithm. Specifically, the Teaching Learning Optimization
Algorithm (TLBO) proposed by Rao et al. (2011) is employed. The details pertaining to the integration of the analytical
model with the optimization algorithm, as well as the estimation of the kinematic variables associated with the downburst
event, are explained in detail in Xhelaj et al. (2022). The TLBO algorithm it is an iterative, stochastic, and population-
based algorithm comprising two distinct phases: the Teacher Phase and the Learner Phase. In the Teacher Phase, the best
solution in the population (the teacher) shares its knowledge (objective function) with the other solutions (the students)
to enhance their performance. In the Learner Phase, the students interact with each other to further improve their
performance. TLBO requires only two user-specified parameters: the maximum number of iterations $T$ and the population
size $N_p$. When incorporating the objective function into a stochastic metaheuristic optimization algorithm, running the
algorithm independently multiple times is crucial to reach the optimal solution. This iterative approach allows for a deeper
exploration of the variable space, reducing the risk of getting trapped in local optima. However, it is important to note
that in the context of metaheuristic optimization, there is no guarantee of attaining a globally optimal solution. As a result,
the procedure can yield a range of solutions ordered based on the values assumed by the objective function, with some
being better than others. In this study, the TLBO algorithm is executed 1024 times independently, with each run producing
an optimal solution. Consequently, 1024 solutions are obtained. The reconstruction of the downburst event can be
accomplished by selecting the solution with the lowest objective function value, as it is considered the best representation



of the event based on the optimization process. This study aims to analyze and clarify the nature of all the solutions
generated by means of the TLBO algorithm for the downburst outflow reconstruction. This choice was made for a twofold
reason.

- • The first reason is to determine the best possible solution among the 1024 totals, where best solution is the one
- that minimizes the objective function $F$, and allows to reconstruct the Sânnicolau Mare downburst event.
- • The second reason, which is the primary objective of this study, is to analyze these 1024 solutions using
- multivariate data analysis (MDA). The method used in MDA are the Agglomerative hierarchical clustering
- (AHC) coupled with the K-Means algorithm and principal component analysis (PCA).

The objective is to investigate the distinct characteristics of the different solutions provided by the TLBO algorithm,
enabling an understanding of their divergence from the optimal solution. If alternative solutions do exist, it signifies that
the algorithm's solution is not unique. As such, a more comprehensive definition of the objective function is necessary to
accurately discern between the optimal solution and its alternatives.
**4.2 Metaheuristic optimization and reconstruction of the Sânnicolau Mare downburst**
In metaheuristic optimization, a commonly used guideline suggests setting the population size $N_p$ as ten times the number
of variables to estimate $D$ (Storn, 1996). In this study, where $D$ corresponds to 11 variables, a population size of $N_p = 110$
has been chosen. Additionally, considering the reported fast convergence rate of the TLBO algorithm (as mentioned in
Xhelaj et al., 2022), the maximum number of iterations $T$ for this study has been set to $T = 100$. Table 2 displays the lower
and upper bounds of the optimization problem pertaining to the reconstruction of the Sânnicolau Mare downburst. These
parameter values have been determined based on a comprehensive literature review, available in Xhelaj et al. (2022).
**Table 2. Lower and upper bound of the decision variable parameters for the reconstruction of the Sânnicolau Mare**
**downburst. Table form Xhelaj et al. (2022).**

| | Parameters/Variables | Lower Bound | Upper Bound |
|---|---|---|---|
| 1 | $X_{C0}$ $(m)$ | -10000 | -10000 |
| 2 | $Y_{C0}$ $(m)$ | -10000 | -10000 |
| 3 | $R$ $(m)$ | 200 | 2000 |
| 4 | $\rho = \dfrac{R_{max}}{R}$ $(-)$ | 1.6 | 2.6 |
| 5 | $V_{r,max}$ $(m/s)$ | 0 | 40 |
| 6 | $T_{max}$ (min) | 2 | 15 |
| 7 | $T_{end}$ (min) | 15 | 60 |
| 8 | $V_t$ $(m/s)$ | 0 | 40 |
| 9 | $\alpha_t$ (deg) | 0 | 359.9 |
| 10 | $V_b$ $(m/s)$ | 0 | 40 |
| 11 | $\alpha_b$ (deg) | 0 | 359.9 |


The spatial domain of the downburst simulation covers an area of 20 x 20 km² while the grid resolution in both the X and
Y directions is set at 50 m. At the center of the domain is placed the probe that sense the time histories of the wind velocity
and direction due to the passage of the simulated downburst. Figure 7 illustrates the "performance chart" depicting the
convergence pattern of the objective functions during the reconstruction of the Sânnicolau Mare downburst using the



TLBO algorithm. The performance chart in Figure 7 illustrates the convergence pattern of the objective functions as
iterations progress. It shows the upper and lower envelopes that encapsulate all 1024 independent runs. The region within
the envelopes represents the objective function values' trend for all runs. At the end of the 100 iterations, the lower
envelope represents to the best objective function value obtained, while the upper envelope corresponds to the worst
objective function value obtained by the TLBO algorithm. The performance chart in Figure 7 includes additional visual
representations: a dashed line representing the mean convergence curve, and dotted lines representing the mean
plus/minus one standard deviation curves. These curves provide insights into the average behavior and deviation of the
objective function values across the 1024 runs. Based on the analysis of the performance charts, it can be observed that
the TLBO algorithm attains convergence after approximately 70 iterations.  At the conclusion of 100 iterations, the best
and worst objective function values correspond to $F_{min} = 0.730$ and $F_{max} = 1.062$, respectively. The mean and standard
deviation of the objective function values are determined as $m_F = 0.893$ and $s_F = 0.080$, respectively.

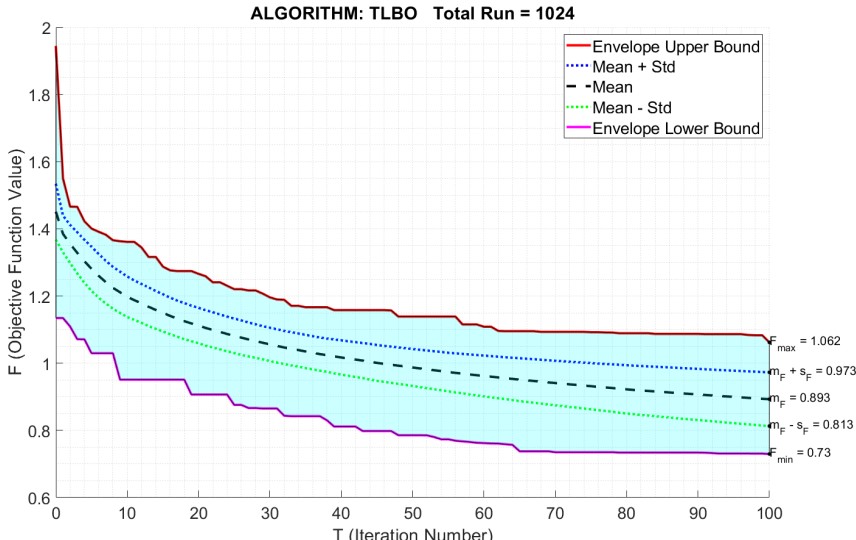


**Figure 7. Performance chart for the reconstruction/simulation of the Sânnicolau Mare downburst using the TLBO algorithm.**

**4.3 Multivariate data analysis of solutions for the Sânnicolau Mare downburst reconstruction**
The optimization algorithm provides in output a data table, where each row of the table is a solution of the optimization
problem. Therefore, the data table is composed of 1024 rows (solutions). The table has 12 columns, where 11 columns
represent the 11 variables/parameters of the analytical model, while the last column contains the values assumed by the
objective function $F$ of each solution (i.e., each row). Although the objective function $F$, is not a variable of the analytical
model, it is treated in Section 5 as a variable from the point of view of the multivariate data analysis. The solutions are
sorted in descending order based on their objective function value $F$. This means that the best overall solution among the
1024, lies in the last row of the data table. The analysis of the data table indicates that most variables exhibit multimodal
histograms, with two or more peaks. However, only the variables $V_b$ and $\alpha_b$ are characterized by a unimodal histogram.
Since the aim of this document is to conduct a multivariate data analysis (MDA), the variables of the data table are split





into primary and secondary variables. Primary variables participate in the analysis of multivariate data (i.e., AHC + K-
Means and PCA), as opposed to secondary variables, which have no role in the calculation. However, secondary variables
can indeed assist in the interpretation of the data table. In the present study, $V_b$, $\alpha_b$ and $\alpha_t$ are considered as secondary
variables. This choice is primarily driven by the observation that $V_b$, and $\alpha_b$ exhibit unimodal histograms, suggesting that
they may not significantly contribute to distinguishing different cluster solutions. However, the choice of $\alpha_t$ as a
secondary variable is purely practical, since it makes it possible to carry out a multivariate statistical analysis, avoiding
the problem of circular statistics and, hence, simplifying the calculation.
Let's define the data table that contains only primary variables by a matrix $\mathbb{X}$. Each row $i$ of the matrix represents a
solution vector $\boldsymbol{X}_i$, encompassing the values associated with the nine primary variables. Therefore the solution vector can
be expressed as $\boldsymbol{X}_i = \left(X_{C0_i}, Y_{C0_i}, R_i, \rho_i, V_{r,max_i}, T_{max_i}, T_{f_i}, V_{t_i}, F_i\right)^T$ with $i$ ranging from 1 to $I$, where I represents the
total number of solutions, in this case $I = 1024$. Since the solution vector $\boldsymbol{X}_i$ contains $K = 9$ primary variables, the
resulting data matrix $\mathbb{X}$ is an $I$-by-$K$ matrix with 1024 rows and 9 columns. For the sake of simplicity, in order to shorten
the notation, let $X_{ik}$ be the value of the $k$-th primary variable in the $i$-th solution. Henceforth, the term "variable" will
refer to primary variables, unless explicitly specified. Consequently, the dataset within the matrix $\mathbb{X}$ can be regarded either
as a collection of rows representing solutions to the optimization problem or as a collection of columns representing
variables of the analytical model. The focus of the MDA lies in examining the data matrix from both the solution and
variable perspectives, aiming to identify similarities among solutions based on their variables. In essence, the goal is to
establish a typology of solutions by identifying groups that exhibit homogeneity in terms of variable similarity. This
analysis allows for a comprehensive understanding of the relationships and patterns among the solutions, facilitating the
identification of distinct solution clusters based on their shared variable characteristics. Since a generic solution $\boldsymbol{X}$, is a
set of $K = 9$ numerical values, $\boldsymbol{X}$ evolves within a space $\mathbb{R}^K$ (a space with 9 dimensions), called "the solution's space".
Defining in the solution's space the usual Euclidean metric (i.e., the $l_2$ norm $\|\cdot\|_2$), then, the squared distance between
two solutions $\boldsymbol{X}_i$ and $\boldsymbol{X}_l$ can be expressed by the Euclidean distance $d_{il}$:

$$d_{il}{}^2 = d^2(\boldsymbol{X}_i, \boldsymbol{X}_l) = \|\boldsymbol{X}_i - \boldsymbol{X}_l\|_2^2 = \sum_{k=1}^{K}(X_{ik} - X_{lk})^2 \qquad (1)$$

The distance $d$ possesses the following metric properties:
$$\begin{cases} d(\widehat{\boldsymbol{X}}_i, \widehat{\boldsymbol{X}}_l) = 0 \iff i = l \\ d(\widehat{\boldsymbol{X}}_i, \widehat{\boldsymbol{X}}_l) = d(\widehat{\boldsymbol{X}}_l, \widehat{\boldsymbol{X}}_i) \quad \text{(simmetry)} \\ d(\widehat{\boldsymbol{X}}_i, \widehat{\boldsymbol{X}}_l) \le d(\widehat{\boldsymbol{X}}_i, \widehat{\boldsymbol{X}}_j) + d(\widehat{\boldsymbol{X}}_j, \widehat{\boldsymbol{X}}_l) \quad \text{(tirangle inequility)} \end{cases}$$
The Euclidean distance not only enables distance calculations but also allows for the definition of angles and,
consequently, orthogonal projections. This concept is fundamental in principal component analysis since PCA relies on
the Euclidean distance as a key component of its methodology. Conducting the analysis directly on the data matrix $\mathbb{X}$
could be misleading without any kind of standardization or normalization. Standardization of the data is essential,
particularly when variables are expressed in different units (refer to Table 1), as it ensures comparability and removes the
influence of scale variations. Additionally, even when variables share the same units, disparities in the range of variability
can skew the analysis. Therefore, normalization operations become crucial to provide equal weight and significance to
each variable, which ultimately leads to a more comprehensive understanding of the data's structure and relationships.
Therefore, in the present work the variables are standardized according to the following equation:



$$\hat{X}_{ik} = \frac{X_{ik} - \bar{X}_k}{S_k}, \qquad \forall i = 1, \dots, I = 1024 \quad and \quad \forall k = 1, \dots K = 9 \tag{2}$$

where $\bar{X}_k$ denotes the sample mean of the $k$-th variable calculated over all $I$ solutions: $\bar{X}_k = \frac{1}{I}\sum_{i=1}^{I} X_{ik}$ and $S_k$ is the
sample standard deviation of $k$-th variable: $S_k = \sqrt{\frac{1}{(I-1)}\sum_{i=1}^{I}(X_{ik} - \bar{X}_k)^2}$.
From a geometric standpoint, the standardization operation holds meaningful interpretations within the solution's space
$\mathbb{R}^K$. The centring operation $X_{ik} - \bar{X}_k$ is equivalent to relocating the origin of the reference system to the centre of mass
of the point cloud. The centre of mass coordinates, $\bar{X}_k$ (for $k = 1, \dots, K$), represent the new origin. The standardization
operation, which consists of considering $\hat{X}_{ik}$ rather than $X_{ik}$, modifies the cloud's shape harmonizing its variability across
all directions. Finally, the normalized data matrix $\hat{\mathbb{X}}$ containing the set of vectors $\hat{X}_i$, $i = 1, \dots, I$, has been used in the
MDA for the identification of different typology of solutions provided by the TLBO algorithm for the
simulation/reconstruction of the Sânnicolau Mare downburst. Figure 8 showcase a summary statistic in the form of a box
plot, illustrating the distribution of the standardized variables. Variables such $\hat{R}_{max}$ and $\hat{T}_{max}$ have a large number of
outliers which indicates extreme values within the dataset. Therefore, even in the context of standardized data, outliers
can still be informative and may hold important information for distinguishing distinct solution clusters.

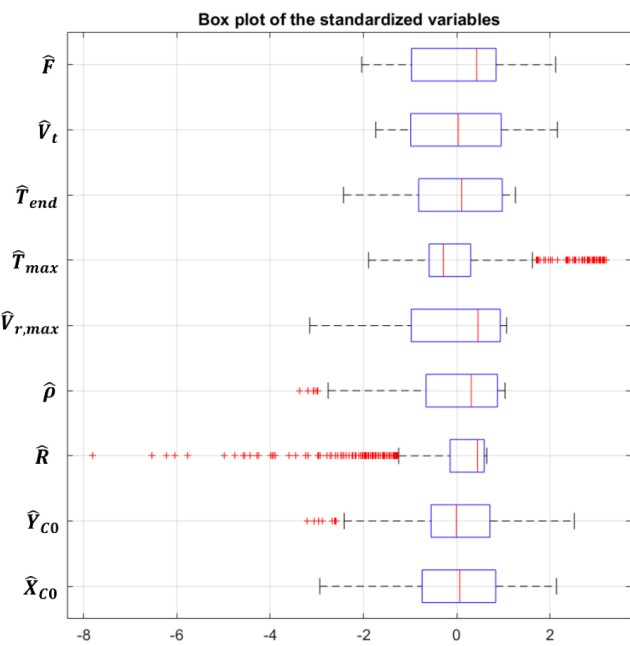

**Figure 8. Box plot of the distributions of the standardized variables. Outliers in the data are plotted individually using the red marker symbol + .**

**5 Results**
In the following section the results of multivariate data analysis (MDA) including cluster analysis and principal
component analysis applied to the data matrix $\hat{\mathbb{X}}$ is presented. After the clusters have been established a comprehensive



description of each of them is provided. This involves examining the variables that contribute to each cluster's composition
as well as identifying specific representative solutions within each cluster. Such an analysis allows for a deeper
understanding of the cluster characteristics and facilitates the interpretation of meaningful patterns and insights within the
data. Sections 5.1 to 5.3 provide an in-depth analysis of data matrix $\mathbb{X}$ from the variable's perspective, employing
agglomerative hierarchical K – means clustering and principal component analysis. In Section 5.4 the clusters are
analyzed from the point of view of the specific solutions which are the most representative of the clusters. Finally, these
representative solutions are compared with the best overall solution founded from the TLBO algorithm. The comparisons
of the representative solution for each cluster and the best overall solution with the full-scale data is therefore enriched
considering the data from the damage campaign that was carried out after the Sânnicolau Mare downburst event.

**5.1 Identification of the most meaningful clusters**

In order to identify the appropriate number of clusters for grouping the solutions, the agglomerative hierarchical clustering
(AHC) is firstly employed (Hartigan, (1975), Kaufman and Rousseuw (1990)). In AHC, each individual solution is
initially treated as an independent cluster (leaf). Through a series of iterative steps, the most similar clusters are
progressively merged, forming a hierarchical tree structure known as a dendrogram. This merging process continues until
all the individual clusters are combined into a single cluster (root). Subsequently, the hierarchical tree is analysed, and a
suitable level is chosen to cut the tree, leading to distinct and meaningful clusters. The number of clusters obtained from
the AHC forms a partition of the data set. To refine and optimize this partition, a partitioning clustering algorithm called
K-means (MacQueen, 1967, Hartigan and Wong, 1979) is subsequently applied. Partitioning algorithms, like K-Means,
subdivides the data sets into distinct clusters, ensuring that solutions within each cluster are similar to one another while
exhibiting noticeable differences between clusters. Hence the two steps clustering procedure is called agglomerative
hierarchical K – means clustering (AHK-MC) and is employed to analyse the standardized data matrix $\mathbb{X}$. By combining
the strengths of both algorithms AHC and K-means, AHK-MC aims to provide a comprehensive and improved clustering
algorithm of the data, enabling a more accurate identification of distinct solution groups. The hierarchical tree (i.e.,
dendrogram) is constructed following the Wards' method (Ward, (1963)). Figure 9 shows the structure of the dendrogram
obtained according to the Wards' algorithm. Since the total solutions of the optimization problem are $I = 1024$, the
dendrogram is very dense at the bottom level (i.e., at the leaf level, where each solution is considered as a cluster by
itself). The hierarchical tree is composed therefore by $I - 1 = 1023$ nodes, the points were two clusters (solutions or set
of solutions) are merged. The level (height) of each node in the tree is described by the within-cluster variance. The level
of a node in the agglomeration process, when examined from top to bottom, indicates the reduction in within-cluster
variance achieved by merging two connected clusters. This reduction in variance can be visualized using a bar graph, as
depicted in Figure 10. From Figure 10 it is possible to establish the level where to cut the dendrogram and consequently
to establish the number of clusters for partitioning the data set. The choice of the number of clusters is important because
partitioning with too few clusters risk leaving groups which are not at all homogeneous. On the other hand, partitioning
with too many clusters' risks creating classes that are not very different from each other. Being $\sum_{s=1}^{I-1} \Delta_s = K = 9$ (the
total variance contained in the standardized data), the separation into two groups is able to describe $\Delta(1,2)/K = $
$4.314/9 = 0.4793$ (47.93 %) of the total variance. Considering the partitioning into three groups, the explained variance
by the three clusters is equal to $[\Delta(1,2) + \Delta(2,3)]/K = [4.314 + 1.044]/9 = 0.5954$ (59.54 %) of the total variance,
while for four clusters the "explained variance" is equal to $[\Delta(1,2) + \Delta(2,3) + \Delta(3,4)]/K = $
$[4.314 + 1.044 + 0.406]/9 = 0.6404$ (64.05 %) of the total variance. Therefore, considering more than three clusters
(refer to Figure 10) is going to have a very little impact on the explained variance since very little information is gained





and is no longer useful to group together any more classes. For this reason, the dendrogram in this work is partitioned in
3 clusters (refer to Figure 10) and therefore they can explain approximately 60% of the total variance present in the data.

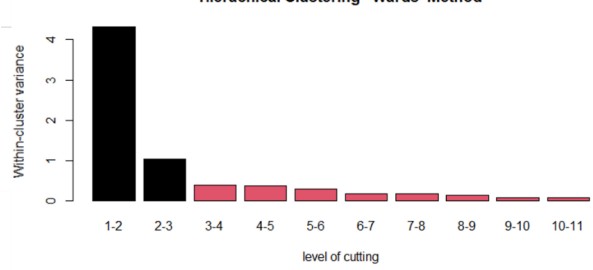

**Figure 9. Hierarchical tree (dendrogram) constructed according to the Wards' Method relative to the solutions of the**

**optimization problem for the Sânnicolau Mare downburst.**

**Figure 10. Bar graph of the relation between the number of merged clusters and the within-cluster variance.**

### 5.2 Interpretation of the clusters through pca and optimization using k-means

The three clusters of solutions are analyzed using principal component analysis (PCA) to identify the key variables that
drive the system's behavior. By extracting the principal components, which captures the most significant variation in the
data, the complexity of the system can be reduced. In particular, the eigenvalues of the correlation matrix $\mathbb{S} = \frac{1}{(I-1)}\hat{\mathbb{X}}^T\hat{\mathbb{X}}$
quantify the amount of variance accounted by each principal component (Kassambara, 2017). The eigenvalues shows that





the first components have larger values, indicating that they capture the most significant variation in the data set. In
contrast, the subsequent components have lower eigenvalues, representing a diminishing level of variation.
Table 3 presents displays the eigenvalues, the percentage of variance explained by each component, and the cumulative
percentage of variance.
**Table 3. PCA results in term of the eigenvalues, percentage of variance and cumulative percentage of variance.**

|  | Dim-1 | Dim-2 | Dim-3 | Dim-4 | Dim-5 | Dim-6 | Dim-7 | Dim-8 | Dim-9 |
|---|---|---|---|---|---|---|---|---|---|
| Eigenvalues ($\lambda_s$) or variance | 5.278 | 1.458 | 0.884 | 0.499 | 0.378 | 0.195 | 0.167 | 0.093 | 0.048 |
| Percentage of variance | 58.645 | 16.204 | 9.825 | 5.542 | 4.204 | 2.170 | 1.852 | 1.028 | 0.530 |
| Cumulative perc. of variance | 58.645 | 74.849 | 84.674 | 90.216 | 94.420 | 96.589 | 98.441 | 99.470 | 100.000 |


The first two principal components capture 74.85% of the total variance in the dataset. These components define a plane
that provides significant insights into the underlying patterns and structure of the data.  Eigenvalues greater than 1 (Table
3) signify that the respective principal components explain more variance in the data compared to any single standardized
variable. These principal components capture significant patterns and structures in the data, contributing more to the
overall variability. In contrast, eigenvalues less than 1, starting from the third principal component (Table 3) indicate that
the associated principal components explain less variance than individual standardized variables, suggesting they have
relatively less influence on the overall variability in the data. Therefore, it is probably not useful to interpret the next
dimensions and better focusing on the first two principal dimensions for a more meaningful analysis.  It is worth
mentioning that the percentage of variance explained by the first principal component (58.65 %) is very close to the
variance explained by the hierarchical tree when is partitioned into three clusters (59.54 %). The three clusters, founded
using the Wards' method only, are represented in terms of solutions in the principal component map (Figure 11a). This
figure shows how solutions are grouped together into three clusters when the overall cloud of solutions is projected into
the first two principal components. Here, cluster 1 is not very well separated from cluster 3, which means that both clusters
share similar solutions. In order to have a better partitioning, the partition is improved (or "consolidated") by applying
the K-Means algorithm to the initial partition (Figure 11a) founded by the Wards' method. Figure 11b shows the principal
component map of the final partitioning of the solutions as result of the application of the K-means algorithm. The
application of the K-Means algorithm improves the partitioning quality since the three clusters this time are very well
separated from each other and are more compact. This final partitioning is therefore maintained and used for the next
analysis of this paper.



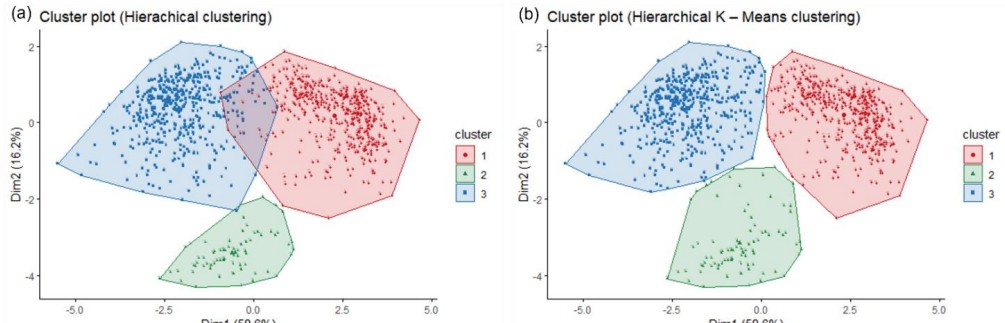

**Figure 11. (a). Solutions' clusters partitioning on the principal component map, using the Ward's method only. (b) Solutions'**
**clusters partitioning using the hierarchical K-means method.**

**5.3 Further considerations on the model's parameters**
Table 4 provides insights into the interpretation of the results from the perspective of the variables, focusing on the first
two principal components. The table displays the correlations between the variables and these components (column 1 and
4). Additionally, it includes the quality of the representation (i.e., projection) of each variable on the first two principal
components (columns 2 and 5), as well as the weight of each variable to the construction of these components (columns
3 and 6) (Husson anf Pagès (2017)). It is important to note that the variables in the Table 4 are vectors which represent
the values observed across the 1024 solutions.
Table 4. Principal component analysis results for variables in terms of correlations, quality of representation and contribution to the
construction relative the first two principal components.

| Variables $\widehat{V}_k$ | Dim-1 $r(\widehat{V}_k, p_1)$ | Dim-1 $qlt_1(\widehat{V}_k)$ | Dim-1 $qtr_1(\widehat{V}_k)$ | Dim-2 $r(\widehat{V}_k, p_2)$ | Dim-2 $qlt_2(\widehat{V}_k)$ | Dim-2 $qtr_2(\widehat{V}_k)$ |
|---|---|---|---|---|---|---|
| $\widehat{X}_{C0}$ | -0.831 | 0.691 | 13.094 | -0.443 | 0.196 | 13.441 |
| $\widehat{Y}_{C0}$ | 0.723 | 0.523 | 9.912 | -0.489 | 0.239 | 16.377 |
| $\widehat{R}$ | 0.578 | 0.334 | 6.326 | -0.256 | 0.066 | 4.504 |
| $\widehat{\rho}$ | 0.715 | 0.512 | 9.699 | -0.216 | 0.047 | 3.200 |
| $\widehat{V}_{r,max}$ | -0.909 | 0.827 | 15.664 | 0.079 | 0.006 | 0.424 |
| $\widehat{T}_{max}$ | 0.182 | 0.033 | 0.628 | 0.916 | 0.839 | 57.502 |
| $\widehat{T}_{end}$ | -0.823 | 0.678 | 12.847 | 0.117 | 0.014 | 0.942 |
| $\widehat{V}_t$ | 0.969 | 0.939 | 17.789 | 0.132 | 0.017 | 1.189 |
| $\widehat{F}$ | 0.861 | 0.741 | 14.042 | 0.188 | 0.035 | 2.421 |
| *Secondary variable* | | | | | | |
| $\widehat{V}_b$ | 0.299 | 0.089 | - | -0.073 | 0.005 | - |


In Table 4, is also present the secondary variable $\widehat{V}_b$. The other two variables $\widehat{\alpha}_t$ and $\widehat{\alpha}_b$ are non-considered since their
interpretation is not consistent with the principal component analysis approach. Despite $\widehat{V}_b$ not being involved in the
construction of the principal components, it is still possible to evaluate the correlation and the quality of the representation



of this variable using the two principal components. To facilitate the interpretation of Table 4, a correlation circle plot
(Abdi and Williams, 2010) can be used to visually represent the variables. This plot represents each variable as a point in
a two-dimensional space, where the coordinates of each point correspond to the correlation coefficients between the
variable and the two principal components (i.e., $r(\widehat{V}_k, p_1), r(\widehat{V}_k, p_2)$). Figure 12a illustrates the correlation circle plot.

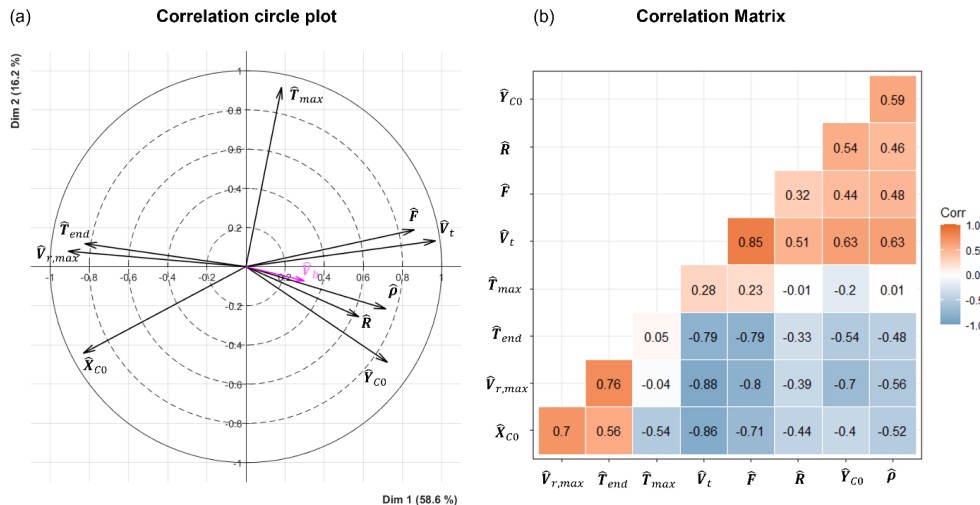

**Figure 12. (a) Correlation circle plot. The variables in black are considered as primary variables whereas the variable in**
**magenta is a secondary variable (b) Correlation matrix plot.**
This plot offers a geometric representation of the correlations among all variable pairs, making it easier to observe the
grouping of positively correlated variables and the positioning of negatively correlated variables on opposite sides relative
to the origin. The total contribution of each variable's representation across all principal components is always equal to
1(i.e., $\sum_{s=1}^{K} qlt_s(V_k) = 1$). If a variable is explained by the first two components, then the sum of its contribution
$\sum_{s=1}^{K=2} qlt_s(V_k)$ will be equal to 1. This implies that the variable's location on the correlation circle will exactly lie on the
circumference of radius 1. Hence, a low-quality variable, which is not very well represented by the first two principal
components will be positioned close to the center of the circle. Therefore, only well represented variables can be
interpreted from the correlation circle. Except for the variables $\widehat{V}_b$ and $\widehat{R}$ which are not very well represented by the 2
principal components, the remaining variables are very well represented since their tip is close to the circle of radius 1.
The set of variables $\{\widehat{V}_t, \widehat{F}, \widehat{Y}_{C0}, \widehat{\rho}\}$ are positively correlated with each other; this means that an increase in one variable
is followed by an increase in the other variable. The same is true for the variables $\{\widehat{V}_{r,max}, \widehat{X}_{C0}, \widehat{T}_{end}\}$. The variable $\widehat{V}_t$ is
highly correlated with the first dimension (correlation of 0.97). This variable could therefore summarize itself the first
principal component axis. From Figure 12a is possible to show that the variable $\widehat{V}_t$ is strongly negatively correlated with
the variables $\{\widehat{V}_{r,max}, \widehat{X}_{C0}, \widehat{T}_{end}\}$. This means for example that solutions which are characterized by high value of storm
motion $V_t$ will systematically be characterized by low values in the maximum radial velocity $V_{r,max}$, "low values" of the
touch down component $X_{C0}$ with respect to the station (which means that for lower positive values of $X_{C0}$, $X_{C0}$ will be
close to the station, while for lower negative values of $X_{C0}$, $X_{C0}$ will be far from the station) and low values of the total
duration of the downburst event $T_{end}$. Since $\widehat{V}_t$ is positively correlated with the variables $\{\widehat{F}, \widehat{Y}_{C0}, \widehat{\rho}\}$, what is true for $\widehat{V}_t$



with respect to the group of variables $\{\widehat{V}_{r,max}, \widehat{X}_{C0}, \widehat{T}_{end}\}$, will also remain true for the variables $\{\widehat{F}, \widehat{Y}_{C0}, \widehat{\rho}\}$. Finally, from
the correlation circle plot, it seems that the variable $\widehat{T}_{max}$ is not very well "linearly" correlated with the groups of variables
$\{\widehat{V}_{r,max}, \widehat{T}_{end}, \widehat{\rho}\}$ since it is nearly orthogonal with these variables. From a quantitative point of view the values of the
correlation coefficients between all the pairs of variables are plotted in Figure 12b. Table 4  shows also the values of the
variable's contribution for the construction of the two principal components (columns 3 and 6 respectively). Also in this
case, it is possible to plot these values to understand which variable contribute the most for building the first two principal
axes.  Figure 13a and Figure 13b show respectively the contribution of the variables expressed in percentage for the
reconstruction of the first two principal components.

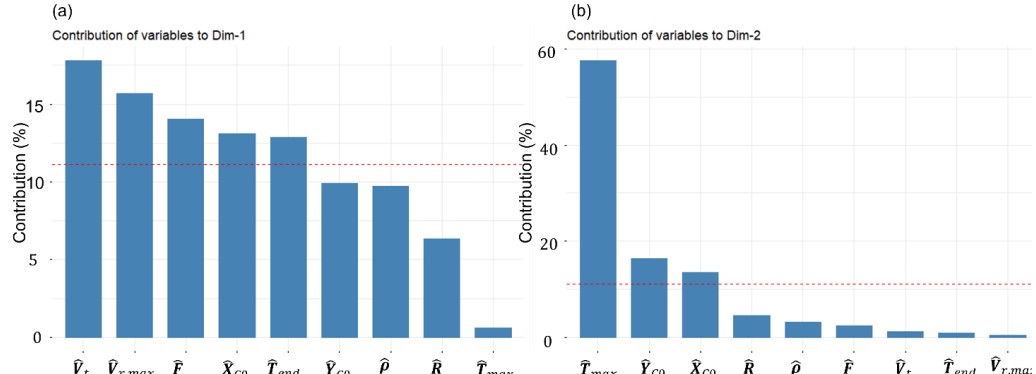

**Figure 13. (a) Contribution of the variables in the reconstruction of the first principal component (Dim-1). (b) Contribution of the variables in the reconstruction of the second principal component (Dim-2).  Variables are sorted from the strongest to the weakest. The red dashed line indicates the expected average contribution.**

The graph displays a red dashed line representing the expected average contribution. If the contribution of variables were
evenly distributed, the expected value would be calculated as 1 divided by the number of variables $K$, which in this case
is 9. This would result in an expected average contribution of 11.11%. For a given component, a variable with a
contribution larger than this cutoff could be considered as important in contributing to the construction of the component.
Therefore, the set of variables $\{\widehat{V}_t, \widehat{V}_{rmax}, \widehat{F}, \widehat{X}_{C0}, \widehat{T}_{end}\}$ contribute the most to the construction of the first principal
component (Dim-1), while the set of variables $\{\widehat{T}_{max}, \widehat{Y}_{C0}, \widehat{X}_{C0}\}$ contributes the most for the construction of the second
principal component (Dim-2). Since the contribution can be added, the set of variables that contributes the most for the
construction of Dim-1 and 2 are given by the set of variables $\{\widehat{V}_t, \widehat{X}_{C0}, \widehat{T}_{max}, \widehat{V}_{rmax}, \widehat{F}, \widehat{Y}_{C0}\}$ which are ordered from the
strongest to the weakest. The remaining variables $\{\widehat{T}_{end}, \widehat{\rho}, \widehat{R}\}$  have a contribution which is lower than the threshold
11.11 %. It is important to observe the partitioning in strongest variables and weakest ones does not represent necessarily
a general case, since the partition might depend on the downburst case under study.

### 5.4 Physical description of the solutions corresponding to clusters 1-3

Once the partitioning of the solutions of the optimization problems in three cluster is completed, it is important to have a
closer look at them and describe common features of solutions which belong to the same cluster. From the partition
analysis, it is found that cluster 1 is made up of 481 solutions, cluster 2 of 85 and cluster 3 of 458 solutions. Table 5
summarizes a few key statistics related to the three clusters. This table includes primary and secondary (i.e., not used for
clustering) variables, which are no longer standardized to investigate their physical meaning.



**Table 5. Description of the partition by the mean and standard deviation of all the variables.**

| Variables $V_k$ | Overall Mean | Overall Std | Cluster 1 Mean | Cluster 1 Std | Cluster 2 Mean | Cluster 2 Std | Cluster 3 Mean | Cluster 3 Std |
|---|---|---|---|---|---|---|---|---|
| $V_t$ (m/s) | 6.025 | 3.371 | 2.811 | 1.042 | 6.527 | 1.407 | 9.307 | 1.492 |
| $X_{C0}$ (m) | -4386.350 | 1613.337 | -3034.079 | 789.682 | -5410.461 | 629.282 | -5616.465 | 1209.346 |
| $T_{max}$ (min) | 6.954 | 2.517 | 5.860 | 1.172 | 13.336 | 1.910 | 6.919 | 1.797 |
| $V_{r,max}$ (m/s) | 24.293 | 5.356 | 28.639 | 1.465 | 28.182 | 1.793 | 19.006 | 3.266 |
| $F$ (-) | 0.893 | 0.080 | 0.823 | 0.058 | 0.919 | 0.043 | 0.961 | 0.021 |
| $Y_{C0}$ (m) | 3363.669 | 1809.316 | 2499.896 | 975.450 | 313.553 | 1257.946 | 4836.890 | 1160.613 |
| $T_{end}$ (min) | 26.035 | 3.167 | 28.269 | 1.895 | 27.622 | 2.295 | 23.394 | 2.235 |
| $\rho$ (-) | 2.189 | 0.108 | 2.126 | 0.104 | 2.134 | 0.100 | 2.265 | 0.050 |
| $R$ (m) | 1334.478 | 102.519 | 1289.518 | 124.661 | 1301.969 | 90.475 | 1387.728 | 23.115 |
| Secondary variables | | | | | | | | |
| $\alpha_t$ (deg) | 290.383 | 0.480 | 276.439 | 0.416 | 253.518 | 0.217 | 310.868 | 0.229 |
| $V_b$ (m/s) | 6.811 | 0.670 | 6.648 | 0.774 | 6.705 | 0.768 | 7.002 | 0.449 |
| $\alpha_b$ (deg) | 268.218 | 0.118 | 264.854 | 0.138 | 273.055 | 0.074 | 270.827 | 0.055 |


In columns 2-3, the overall mean, and the overall standard deviation (std) are calculated with respect to each variable
(primary and secondary). In the other columns, the same calculation was repeated taking into consideration the three
clusters. Mean and the std of the secondary variables $\alpha_t$ and $\alpha_b$ have been calculated using circular statistics (Rao and
Sengupta, 2001). To start clarifying the characteristics of the different clusters, Figure 14 shows the scatter plot and
distribution of the touchdown components $(X_{C0}, Y_{C0})$ for all the solutions, partitioned into three clusters. In this figure it
is shown the center (namely the mean) of each cluster and the location of the touchdown position of the best overall
solution. The figure shows also with a black line the position of the city of Sânnicolau Mare. Also, on the left and on the
top of this figure is possible to show the histograms of the variable $(X_{C0}, Y_{C0})$ relative to each cluster.



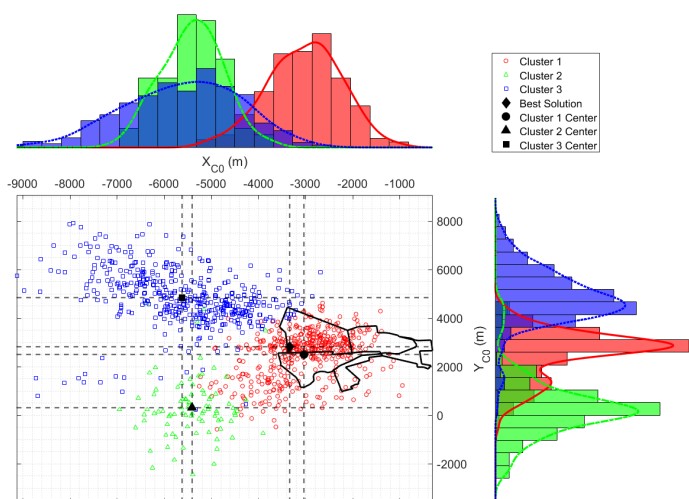

**Figure 14. Scatter plot and histogram density distribution for the variables ($X_{C0}, Y_{C0}$). The dark black line shows the contours of the city Sânnicolau Mare.**

The three clusters appear well separated in terms of touchdown position ($X_{C0}, Y_{C0}$). Since it is very unlikely that the cluster means coincide with one of the solutions present in the data set, let's define as "cluster solution", the solution which is the closest to the mean of the cluster. Accordingly, the cluster solutions, reported in Table 6, will be used to interpret the average features of each cluster. The first row of this table is dedicated to the best solution founded by the optimization algorithm (i.e., the one that have the lowest objective function $F$ among all the solutions); the best solution belongs to Cluster 1.

**Table 6. Overall best solution and clusters representative solutions.**

| Solutions | $V_t$ (m/s) | $X_{C0}$ (m) | $T_{max}$ (min) | $V_{r,max}$ (m/s) | $F$ (-) | $Y_{C0}$ (m) | $T_{end}$ (min) | $\rho$ (-) | $R$ (m) | $\alpha_t$ (deg) | $V_b$ (m/s) | $\alpha_b$ (deg) |
|---|---|---|---|---|---|---|---|---|---|---|---|---|
| Best solution | 2.76 | -3339.53 | 6.50 | 29.80 | 0.73 | 2826.55 | 29.89 | 2.15 | 1381.38 | 271.74 | 5.49 | 58.35 |
| Cluster 1 | 2.51 | -2944.15 | 6.05 | 29.54 | 0.81 | 2769.36 | 27.23 | 2.09 | 1287.53 | 278.25 | 7.15 | 268.19 |
| Cluster 2 | 6.14 | -5105.66 | 14.05 | 27.07 | 0.91 | 383.39 | 28.18 | 2.14 | 1295.33 | 255.36 | 7.13 | 272.82 |
| Cluster 3 | 9.25 | -5930.81 | 7.15 | 17.36 | 0.97 | 4575.50 | 22.95 | 2.27 | 1392.86 | 307.61 | 6.15 | 272.71 |

Figure 15 shows the time histories produced by the best solution and the three cluster solutions, in terms of wind velocity (Figure 15a) and direction (Figure 15b), compared with the moving averaged recorded data. The figure provides a qualitative representation of the goodness of fit between the simulations and the recorded data. The goodness of fit is quantitatively measured by the objective function $F$. The simulations produced from the best solution and the Cluster 1 solution fit the data better than Cluster 2 and 3. This is quite obvious since the best solution have the lowest objective




function $F$ and belongs to Cluster 1, whereas Cluster 2 and Cluster 3 solutions have slightly higher objective function
values $F$ (refer to column 5 in Table 6).

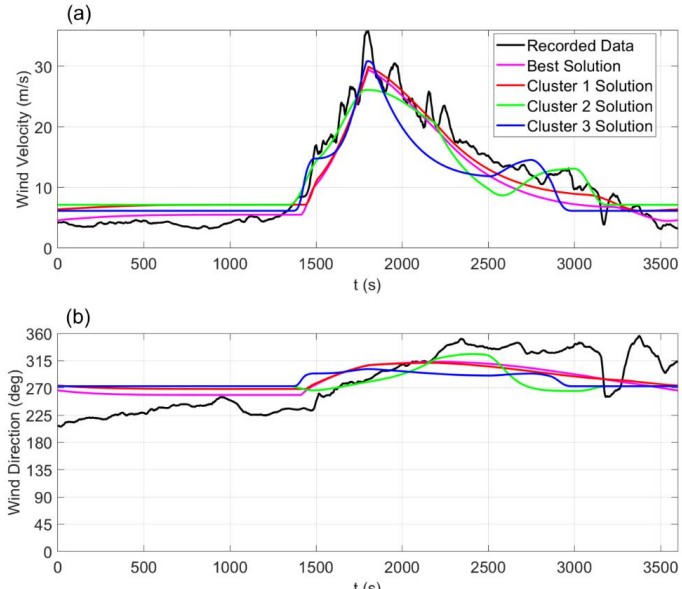


**Figure 15. Comparison among the moving averaged wind speed (a) and direction (b) obtained from the measurements of the**
**Sânnicolau Mare downburst, along with the best solution and the three cluster solutions.**
In order to better understand the nature of the different solutions relative to each cluster, for each solution present in Table
6, the downburst 2D wind velocity is evaluated at the same height of the anemometric station (i.e., at 50 m AGL). The
left panels of Figure 16 (from (a) to (d)) show for each of the 4 solutions the wind filed reconstruction during the
intensification stage of the downburst, while the right panels (from (e) to (h)) describes the stage of maximum intensity.
Note that the time of maximum intensity is different for each cluster according to the corresponding value of $T_{max}$
reported in column 3 of Table 6.



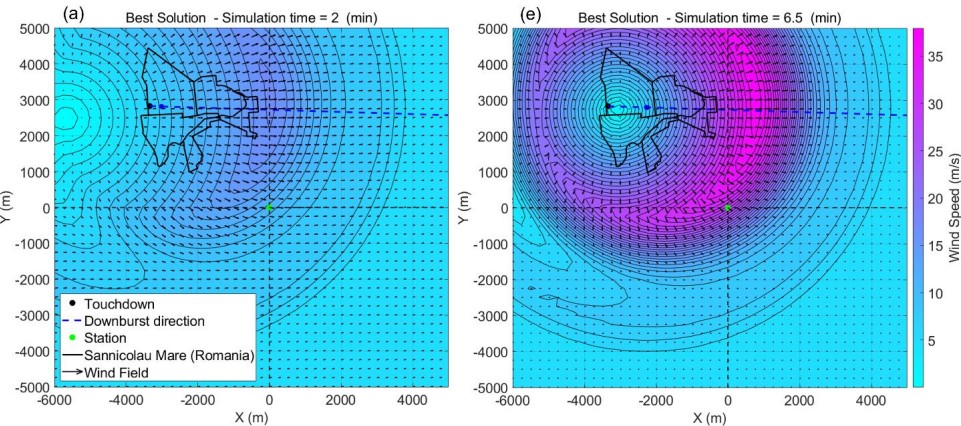


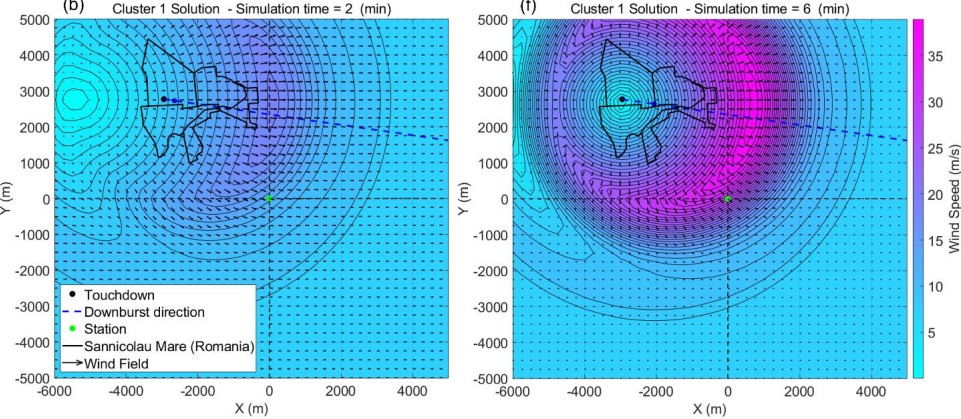


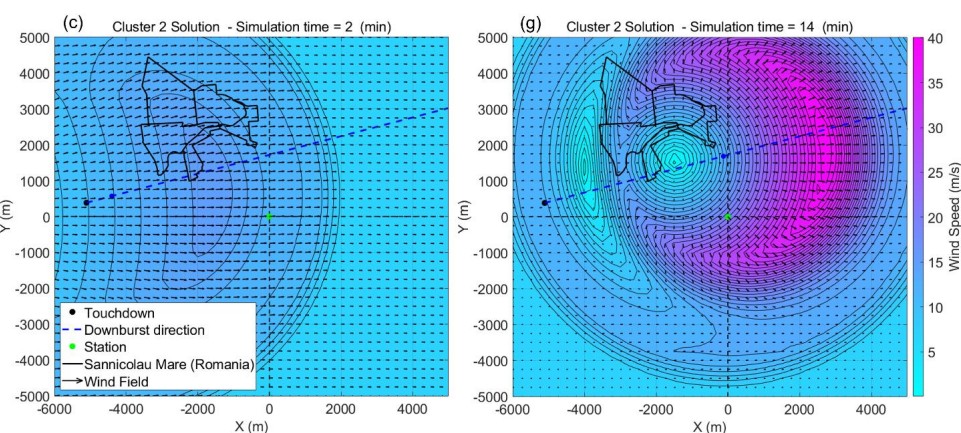






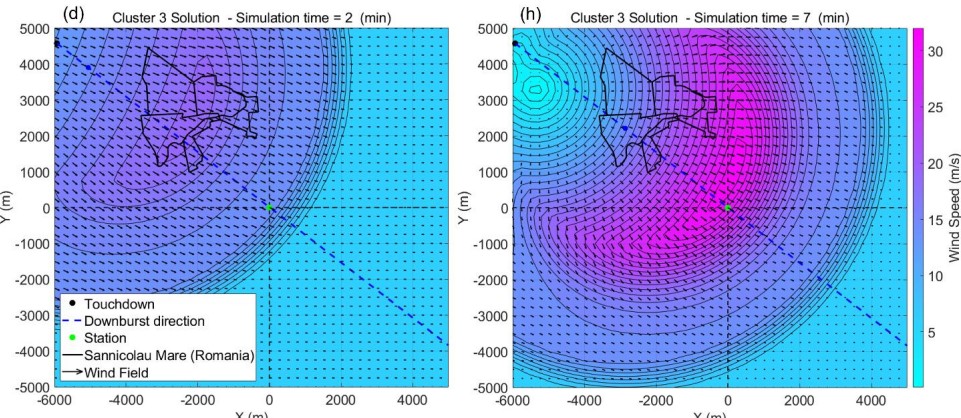


**Figure 16. 2D wind field reconstruction at 50 m AGL. From (a) to (d), the best solution, Cluster 1, 2 and 3 solutions are depicted**
**at the intensification stage of the downburst. From (e) to (h), the best solution, Cluster 1, 2 and 3 solutions are depicted at the**
**maximum intensification stage of the downburst.**

Cluster 1 touches down very close to the city center and moves slowly eastward, it is characterized by a low value of the downburst translation velocity $V_t$, with mean value 2.8 m/s against the overall mean among all clusters which is 6.0 m/s. In addition, it has maximum radial velocity $V_{r,max}$ higher and overall duration of the downburst event $T_{end}$ longer with respect to the mean values of the other two clusters. The solutions belonging to Cluster 2 touch down around 2 km southwest of the city, they propagate northeastward with higher translation velocities compared to Cluster 1 and the longest intensification periods $T_{max}$ overall. The solutions in Cluster 3 touch down about 3 km northwest of the city, they move southeastward with the highest values of downburst translation velocity $V_t$ but they are the lowest-lasting as the duration of the downburst event $T_{end}$ is on average 23.4 min while the overall mean is 26.0 min. They also have the lowest values of maximum radial velocity $V_{r,max}$ which compensate the high translation velocities. According to these descriptions, it is clear that in the solution's space of the model three different solutions exist that can describe similarly the time-series measured at TM_424. The existence of different plausible solutions means that the problem of finding the downburst wind field time-space evolution using a single time-series is an underdetermined problem.

The Sânnicolau Mare downburst had a strong impact, causing hail damage to numerous buildings in the town. A damage survey was conducted to assess the affected areas and identify buildings that experienced hail damage during the event. To estimate the extent of the damage, the simulated wind field generated by the analytical model was utilized. By analyzing the wind speeds at various locations, the "footprint" of the simulated damage was determined. This footprint represents the maximum wind speed recorded at different places during the downburst, providing valuable information on the areas most affected by the event. The left panels of Figure 17, labeled from (a) to (d), depict the complete footprint of the downburst potential damage area for the best solution and the three cluster solutions. In contrast, the right panels, labeled from (e) to (h), provide a closer view of the footprints overlaying the simulated maximum wind velocity vectors (indicated by blue arrows) onto the locations of hail damage. The hail damage is represented by vectors pointing orthogonally to the damaged facades (represented by pink arrows). The comparison between the facades damage, which is related to the trajectory of hail transported by the strong downburst-related outflow, and the simulated maximum



velocity reveals interesting findings. Specifically, the best solution and Cluster 1 solutions exhibit the strongest alignment
between the maximum wind velocity vectors and hail damage vectors, particularly in the central part of the city and along
the path of the downburst. In contrast, Cluster 2 and Cluster 3 demonstrate a consistent deviation of the maximum velocity,
with Cluster 2 deviating northward and Cluster 3 deviating southward, relative to the hail trajectories. This observation
suggests that the actual downburst event likely followed a pattern more closely resembling Cluster 1 rather than the other
two potential solutions.

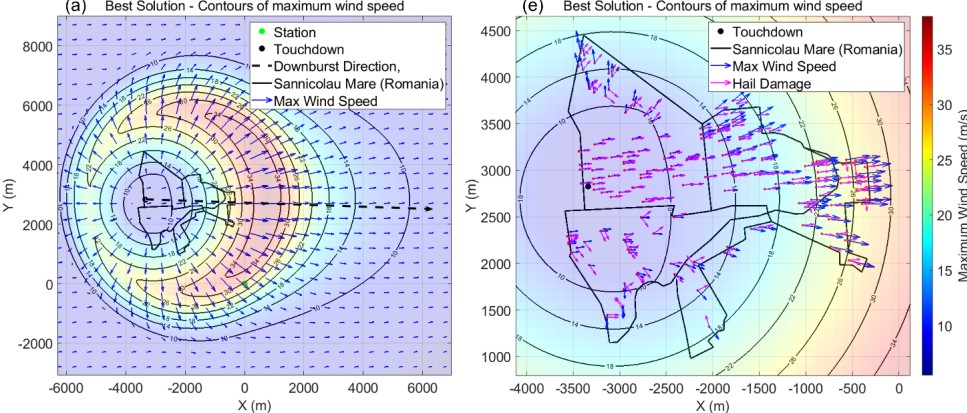


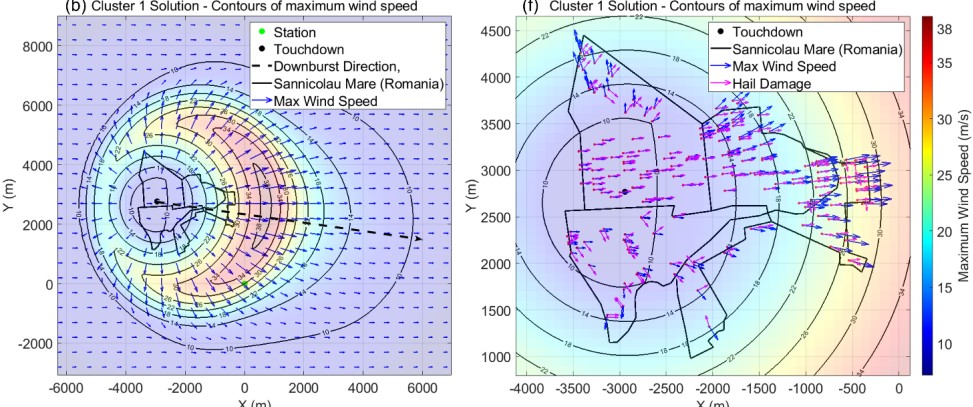






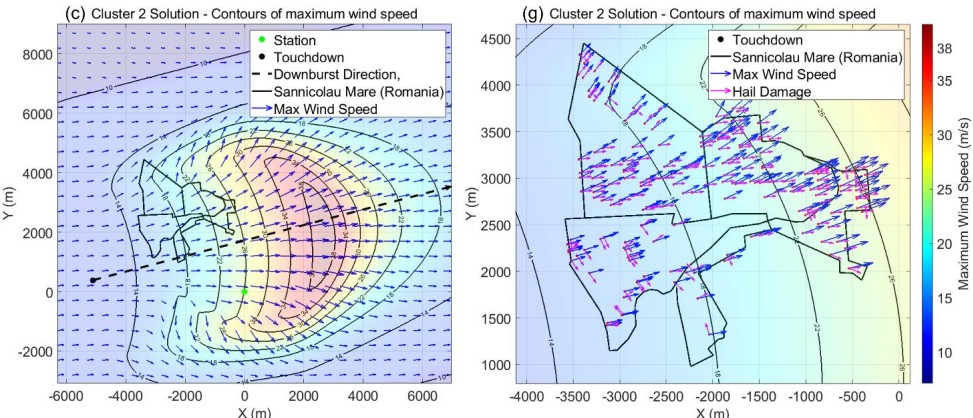


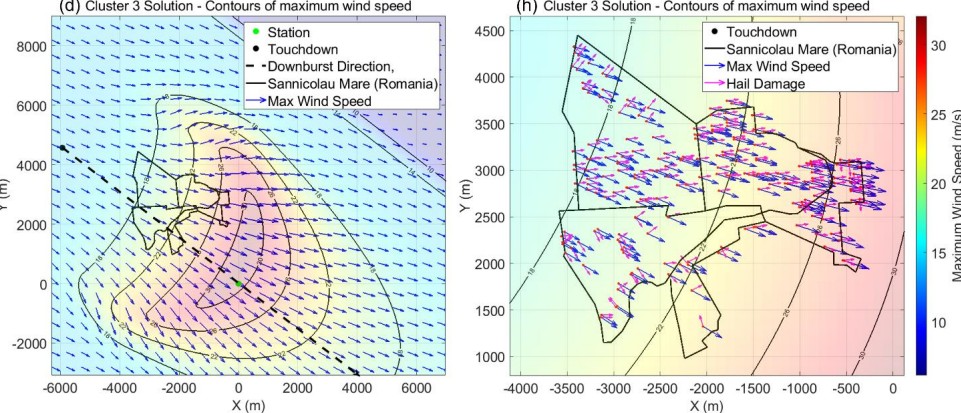


**Figure 17. Simulated footprints of the downburst that occurred in Sânnicolau Mare. Panels from (a) to (d) representing the footprints for the best solution, Cluster 1, Cluster 2, and Cluster 3 respectively. Panels form (e) to (h), representing comparison between hail damage and maximum simulated wind speed for the best solution, Cluster 1, Cluster 2, and Cluster 3 respectively.**

These observations lead to the conclusion that the optimal (best) solution, which minimizes the objective function *F*, is the most reliable among all possible solutions. This implies that the reconstruction of the downburst wind field should be based on a large number of simulations to ensure that the optimal solution is obtained. By conducting numerous simulations, the likelihood of obtaining the most accurate representation of the downburst event is maximized leading to a more accurate reconstruction of the event.





**7 Conclusions**


This study focuses on the analysis of solutions obtained by combining an analytical model (Xhelaj et al., 2020) with a
global metaheuristic optimization algorithm for the reconstruction of the wind field generated during the Sânnicolau Mare
downburst event in Romania on June 25, 2021. The analytical model and optimization algorithm are coupled using the
Teaching Learning Optimization Algorithm (TLBO) to estimate the kinematic parameters of the downburst outflow. The
procedure for this coupling and parameter estimation is described in detail in the study by Xhelaj et al. (2022). Therefore,
the objective was to analyse the differences among the solutions provided by the optimization algorithm and to assess
their physical validity as alternatives to the optimal solution. In the presence of multiple physically sounding solutions, it
has been demonstrated that additional data describing the downburst thunderstorm event is necessary to determine which
solution best represents reality. To support the analysis a comprehensive damage survey was conducted in collaboration
with the University of Genoa (Italy) and the University of Bucharest (Romania) to assess the extent and location of hail
damage on buildings in the affected area. This survey, along with the wind speed and direction signals recorded during
the downburst event by a telecommunication tower located approximately 1 km from the city, significantly enhances the
information available for the reconstruction and simulation of the downburst using the optimization procedure. The
analysis of the solutions generated by the optimization algorithm involves multivariate data analysis (MDA) techniques,
specifically agglomerative hierarchical clustering coupled with the K-means algorithm (AHK-MC) and principal
component analysis (PCA). The AHK-MC is used for classifying the solutions into different clusters based on their
features, while PCA is employed to determine the importance of the variables in the analytical model for the downburst
event reconstruction.
The application of AHK-MC resulted in the identification of three main clusters, each with distinct characteristics, among
the 1024 solutions.
- Solutions belonging to Cluster 1 are characterized by a slow storm motion, small touch down distance from the
city of Sânnicolau Mare and by long duration of the downburst event. The best overall solution belongs to Cluster

574 1.

- Solutions belonging to Cluster 2 are characterized by a moderate storm motion and moderate distance of the
touch-down from the town of Sânnicolau Mare. These solutions are also characterized by high duration of the
intensification period of the downburst event.
- Solutions belonging to Cluster 3 are characterized by a high storm motion and high distance of the touch-down
from Sânnicolau Mare. They are also characterized by low duration of the downburst event and low values of
the maximum radial velocity.
The result of the MDA allows also to establish at least for the case under consideration that the set of variables
$\{V_t, X_{C0}, T_{max}, V_{rmax}, F, Y_{C0}\}$ which are ordered from the strongest to the weakest are the more important for the
reconstruction/simulation of the downburst event. The remaining variables $\{T_{end}, \rho, R\}$ have a lower contribution. It is
important to observe the partitioning in strongest variables and weakest ones does not represent a general case, since the
partition depends on the downburst case under study.
Finally, the comparison between the facades damage, which are related to the trajectory of hails transported by the strong
downburst-related outflow and the simulated maximum velocity shows that the best solution and Cluster 1 solutions seem
to have a "good" overlapping between maximum wind velocity vectors and hail damage vectors. Considering the solutions
of Cluster 2 and 3, it seems that the match between maximum wind velocity vectors gradually decreases, with the worst



case represented by Cluster 3 solutions. These observations allow to conclude that the optimal solution, that is, the one
that minimizes the objective function $F$, is the best with respect to the other three cluster solutions, also from the point of
view of the damage analysis. As a result, for the specific case being examined, relying on the best overall solution provided
by the optimization algorithm appears to yield promising results for reconstructing the downburst wind field. Obviously,
an analysis of this type, conducted on several downburst events, will be able to better confirm this statement.
**Author contributions**
This paper is based on the Ph.D. thesis of Andi Xhelaj, under the guidance of Prof. Giovanni Solari and Prof. Massimiliano
Burlando. Andi Xhelaj played a crucial role in conceptualizing the study, developing the methodology, organizing the
data, conducting data analysis, and preparing the manuscripts and figures. The study was supervised by Prof.
Massimiliano Burlando, who provided guidance and conducted internal review process.
**Declaration of competing interest**
The authors affirm that they have no known financial conflicts of interest or personal relationships that could have
influenced the findings presented in this paper.
**Acknowledgments**
The authors would like to acknowledge the valuable contributions of I. Calotescu, X. Li, M.T. Mengistu, and M.P. Repetto
for providing the time histories of the recorded data and the hail damage map from the survey of the Thunderstorm event
in Sânnicolau Mare, Romania, on 25 June 2021.  The monitoring system and damage survey were carried out as part of
a research collaboration between the University of Genoa (UniGe) and the Technical University of Civil Engineering
Bucharest (UTCB), funded by the European Research Council (ERC) under the European Union's Horizon 2020 research
and innovation program (Grant Agreement No. 741273) for the Project THUNDERR - Detection, simulation, modeling,
and loading of thunderstorm outflows to design wind-safer and cost-efficient structures, supported by an Advanced Grant
(AdG) 2016. As the corresponding author is not a native English speaker, he utilized the OpenAI's GPT-4 model as an
editing tool during the creation of this paper to review and amend grammatical and spelling mistakes and to ensure
linguistic consistency and coherence throughout this paper.

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
