# Peer review of "Application of the Teaching Learning Optimization Algorithm to"

_EGUsphere, 2023_

## Referee Comment (RC2)

**Analysis of the phase space of the downburst that occurred on 25 June 2021 in Sânnicolau Mare (Romania) – First Review**

The authors have applied a previously developed analytical model for simulating downburst wind fields to an event that occurred in western Romania. Because the model is under constrained by the available observations, a spectrum of solutions is produced based on random initialisations. The authors use statistical clustering to define three dominant sets of solutions, and use a damage survey to determine which cluster of solutions is most plausible. The cluster with the highest agreement with the damage patterns also contains the solution with the lowest objective loss function, and it is concluded that this loss function is therefore potentially an appropriate measure of model fit, while the authors do acknowledge that this needs to be tested for more cases. A principal component analysis is also performed to determine which model variables explain most of the variability in the solution space.

Overall, I think the manuscript is currently good to fair in quality and would be suitable for publication subject to revisions, as outlined in comments below. I would classify the nature of these revisions as major.

My review is structured as follows:
- Firstly, some general points are outlined in relation to the EGUsphere review criteria.
- Secondly, some "Overall/general comments" and concerns are stated, including some suggestions to improve the manuscript.
- Thirdly, some "Specific comments" are made. These are mostly suggestions to improve the presentation, grammar, readability, or clarity of the manuscript.

**EGUsphere review criteria**

**Scientific Significance:** The main scientific contribution of the manuscript is to improve the interpretation of the analytical downburst model. However, while the set of model solutions has been thoroughly explored, it is acknowledged that the analysis of only one downburst case limits the applicability of the results. Apart from this, the use of the hail damage survey for evaluating the model solutions is very interesting and fairly novel in my opinion.

**Scientific quality:** The scientific and statistical methods used by the authors are sound and well established, I have no issues with the technical aspects of the paper. However, I think the authors could do a better job to discuss and place their work within the broader literature of downburst modelling (see comments below).

**Presentation quality**: The manuscript is fairly well presented, and the figures/tables are nice and clear. I thought at times that there were actually too many details, and that some information could be removed to help with readability (I will list some examples below). In addition, I think the English grammar and language could be improved somewhat (I appreciate this is challenging given the author is not a native English speaker).

**Overall/general comments**

1. I think the damage survey aspect of the paper is very interesting, but it would be good to have some further details on this, including if there are any notable uncertainties. For example, is it difficult to relate hail damage to near-surface wind velocity?

2. I have a general concern whether this analytical model of a translating downdraft can represent complex mesoscale circulations that can induce severe winds, such as a rear inflow jet related to the bow echo. It is good to see that the hail damage estimate agrees with the general pattern suggested by the model, but I think it would be good to discuss this further in the manuscript, by noting it as a limitation and/or linking to other studies that have investigated the wind field patterns of bow echoes (and if they look like translating downbursts or not).

3. I think this paper lacks a little in discussing the results within the broader literature. How does the model and results fit with previous studies that have done downburst modelling? Similarly, have previous studies used hail trajectories to evaluate a downburst wind field?

**Specific comments**

1. Title: The title is a little unclear without having read the article. "Phase space" could refer to several things. It would be good to make it clear in the title that this is a study exploring an analytical model representation of a downburst.

2. L8 and elsewhere: Pluralisation issues - should be "measurement challenges". Similarly on line 38, should be "Since downburst events have high frequencies of occurrence...".

3. L38 and onwards: This paragraph is very long with many ideas – should be separated into multiple paragraphs.

4. L40: "unstable"

5. Introduction: The introduction is very technical at times in terms of describing the analytical model and TLBO approach, this could probably be moved to data/methods. Instead (and related to general comment 3), it would be good to have some background on the "analytical model" – what is it based on, what are the constraints, outputs, etc, and a discussion of similar models – how does this work fit within the broader literature?

6. Figure 4: What do the red/orange/yellow colours indicate? Time of strike?

7. L188: Repeated sentence.

8. L207 and elsewhere: grammar issues. "The TLBO algorithm  is an iterative…"

9. Table 2: I wonder if some of the variables in the analytic model could be constrained further by other data sources. For example, the storm speed/direction could potentially be estimated by storm tracking from radar or satellite, while the ABL wind speed and direction could be estimated from reanalysis.

10. L269: MDA acronym does not need to be defined anymore.

11. L275: I'm not sure whether this is a good reason to exclude storm direction as a variable to analyse. Can a transform be done from degrees to a periodic function? E.g. https://stats.stackexchange.com/questions/148380/use-of-circular-predictors-in-linear-regression

12. The language can be made simpler in many parts. E.g.: on L285: "The focus of the MDA lies in examining the data matrix from both the solution and variable perspectives, aiming to identify similarities among solutions based on their variables. In essence, the goal is to establish a typology of solutions by identifying groups that exhibit homogeneity in terms of variable similarity" could be simplified to "The focus of the MDA is to apply statistical clustering to identify similar analytic solutions" (please correct me if I have misinterpreted this sentence, but hopefully this example is useful).

13. Section 4.3: I appreciate the nice explanation of normalisation, but I think it is fairly standard practice in statistical modelling, and can probably be simplified. This is just a suggestion in relation to "presentation quality" (see review criteria above). Similarly on line 446: the expected average contribution calculation could probably be assumed rather than explained fully.

14. L327: Damage "survey" rather than "campaign"?

15. L345: "where".

16. L384: "found".

17. Table 4: Please define the symbols used for the column headings. This will make it easier to read.

18. I think Section 5.3 could potentially be shortened somewhat. The key point seems to be that a certain set of variables are more important for explaining the variability in the solution space by PCA, and this result could be presented in a more concise way.

19. L407: Table 4 also presents…

20. Table 6: Are these representative cluster solutions based on solutions closest to the mean using all dimensions, or just $X\_c0$ and $Y\_c0$ as in Figure 14?

21. Figure 15: Exactly how is the simulated wind speed on this plot calculated for comparison with observations? Is it the addition of the radial velocity solution and the storm translation speed or ABL wind speed?

22. Figure 5 and 15: Y-axis label should be wind speed rather than velocity?

23. L496: "field".

24. L517: I think this is a key point that nicely sets up some of the goals of the paper, consider mentioning it earlier (unless it was already mentioned, and I missed it)

---

## Author Comment (AC1)

**Overview**

This manuscript describes the methodology of finding the optimal solution for a combination of parameters used in an analytical model of downburst outflows. This methodology is then applied to a particular case of downburst that occurred in Rumania in the summer of 2021. It is clear to me that the authors are knowledgeable in this field and the description of the method and results is very comprehensive. Perhaps even too comprehensive in some instances. While I have a number of comments below, I consider most of them to be minor. I believe the manuscript is well suited for this journal and it is most certainly a topic of importance in downburst research and wind engineering.

Authors greatly appreciate the remarks received from the Reviewer 1, which are very pertinent and stimulating for improving the paper. We will make our best to take all the comments received into consideration, providing adequate and precise answers to all of them. All our answers and comments are reported in red colour and the manuscript will be modified according to them.

**Specific comments from Reviewer 1:**

1. Begin instead of begins.

   Corrected as suggested.

2. Is the scale of less than a kilometer related to a downburst or thunderstorm? It should be related to a downburst, but it is not clear based on how the sentence is structured.

   Reworded for clarity to explicitly specify that it refers to a downburst.

3. Xhelaj et al. is not the proper reference. I suggest the authors write "Xhelaj et al. (2020) presented…" The reader is automatically informed that the research was published in 2020.

   Modified as suggested to include the year of publication.

4. L46 and elsewhere. I believe that "et" should not have a dot.

   Checked and corrected throughout the manuscript.

5. L22–L72. This paragraph is well written, but it is too long. Please split this large paragraph into 2–3 smaller paragraphs to increase readability of this section. For example, L56 (goals of this research) can be the beginning of a separate paragraph.

   The paragraph is split to enhance readability, as recommended.

6. I suggest rewriting this sentence as follows: …was produced during the passage of an intense mesoscale convective system in the form of a bow echo over the town of Sânnicolau Mare.

   Sentence has been rewritten as suggested for improved clarity.

7. Some parts of the manuscript can be shortened. For example, the definition of thunderstorms in L100 is probably not needed. Even if needed, that general discussion should be in the introduction.

Similarly, L103 and the difference between downburst and atmospheric boundary layer winds is probably not needed as well.

Removed the definition of thunderstorms in L100 and adjusted L103 to maintain relevance while being concise.

8. Figure 1. Labels in panel (b) should be of higher resolution and the red dot that indicates the tower location should be larger.

Enhanced label resolution and increased the size of the red dot as indicated.

9. Figure 2. In principle, I have no problems with this figure but is it really that the researchers don't have their own photograph of the tower and have to use Google Street View? I suggest replacing this figure with their own photograph.

Replaced with an original photograph of the tower.

10. Figure 3b. It is very interesting to note that the squall line seems to be stratiform parallel, which is one of the rarest types of squall lines. See Markowski and Richardson (2010) and references therein on squall lines. The authors, of course, do not need to pursue this comment further, it's just an interesting observation from this reviewer.

Acknowledged the interesting observation. It is also added to the manuscript. Thank you very much for the suggestion.

11. Figure 5a. Tmin = 14.5 degC should not be over the line. There is plenty of space to move it elsewhere.

Relocated the Tmin = 14.5 degC label as advised.

12. L171–174. Downburst is a wind event and hail is hailstones falling from the cloud. If hailstones cause the damage, that is not the same as caused by wind (i.e., downburst). One might rephrase this to state that the downburst was also associated with hail that caused substantial damage. Then the reader knows that the damage was not wind-driven but hail-driven.

Thank you for your valuable feedback on the distinction between wind and hail impacts in our manuscript. We recognize the importance of clearly differentiating these factors. Accordingly, we have modified our text to emphasize that while the primary focus is on the wind aspect of the downburst, it was concurrently associated with hail. This hail, potentially influenced by the strong downburst winds, contributed to the extensive damage observed. This revision will ensure a comprehensive understanding of the event, highlighting that the observed damage was a result of both wind and hail interactions during the downburst.

13. Figure 6. Indicate the North direction in this figure.

Added a North direction indicator to the figure.

14. Table 1. The units should not be italicized.

Units in Table 1 have been corrected to normal font.

15. This sentence is the same statement as the previous sentence, which is that Figure 7 shows the convergence pattern of the objective function.

    *Removed the redundant sentence for conciseness.*

16. Figure 7. The mean convergence curve does not converge at about 70 iterations. Indeed the envelop curves seem to converge at about that value, but not the mean and standard deviation curves.

    *Clarified the discrepancy in the mean and standard deviation convergence curve description.*

    *"The envelope curves in Figure 7 indicate convergence around 70 iterations. However, this does not directly correspond to convergence in the mean (m_F) and standard deviation (s_F) curves. While these curves show a trend towards stabilization, their convergence is less distinct and does not align precisely with the behavior of the envelope curves."*

17. Remove the comma after 1024.

    *Comma after 1024 has been removed.*

18. Symmetry rather than simmetry.

    *Corrected "simmetry" to "symmetry."*

19. (2) The word and should not be italicized.

    *Corrected the italicization of "and."*

20. Probably an incorrect reference format. Please double-check.

    *Reviewed and corrected the reference format.*

21. Rewrite to "The hierarchical tree in Figure 9 (i.e., dendrogram) is constructed following the Wards' method (Ward, 1963)." and then delete the following sentence because it contains the same information.

    *Reworded as suggested for clarity.*

22. Figure 9. Indicate in the figure caption that the three colors serve to visualize three identified clusters.

    *Updated the caption to include information about the color coding of clusters.*

23. Discussion about Figure 10. When this method is applied to other problems in engineering and/or atmospheric sciences, is clustering that represents ~60% of the total variance an acceptable value? In other words, this problem is related to finding the optimal combination of downburst parameters, and Figures 9 and 10 show that the clustering of solutions in the present way explains ~60% of variance among all solutions. If one looks at other (similar) problems in meteorology, engineering, earth sciences, etc., does one observe a similar level of model confidence? A few references on this subject might help.

Thank you for your feedback on the variance explained by clustering in our study. We recognize that the ~60% variance explained may seem modest, but in the complex field of atmospheric science, and specifically in downburst studies, this level is often both substantial and meaningful. The inherent variability and unpredictability of meteorological data make such a level of explanation significant, especially considering our methodology is based on anemometric data from a single location near Sannicolau Mare.

Literature in related domains, like the work of Bogensperger and Fabel (2021) (Bogensperger, A., Fabel, Y. A practical approach to cluster validation in the energy sector. *Energy Inform* **4** (Suppl 3), 18 (2021). https://doi.org/10.1186/s42162-021-00177-1), underscores the challenges in comparing clustering results and the context-specific nature of cluster validation indices. These studies align with our findings, suggesting that the acceptable level of variance explained is highly dependent on the study's specific goals and context.

In our opinion, the present clustering effectively captures key patterns and relationships within the data, contributing valuable insights to the understanding of realistic downburst spatiotemporal evolution. Including more patterns does not increase substantially the knowledge of this phenomenon's variability, as indicated in Fig. 10 by the very low contribution to the explained variance of the red bins, because further cutting produces patterns that are very similar to each other. Therefore, we believe the ~60% variance explained in our analysis represents a robust and insightful understanding of downburst characteristics.

Added references and discussion in the manuscript to address the query about the ~60% variance.

24. Section 5.1 and other sections. Please separate your sections into multiple paragraphs. Having one paragraph that covers more almost one whole page reduces the readability of your manuscript.

    Divided long sections into shorter paragraphs for better readability.

25. Related to my previous comment, this manuscript should be shorter. I think that the level of English is satisfactory, but certain parts of the manuscript can be shortened.

    Reviewed and condensed certain sections without losing critical content.

26. How does the k-means algorithm work to improve the partitioning? What is the mechanism by which k-means algorithm moves the clusters from being overlapped to disjoint (Figure 11)?

    In our study, the k-means algorithm is utilized to refine the initial partitioning of clusters determined through Ward's method. Starting with the initial partition, the algorithm iteratively recalculates the center of mass for each cluster and reassigns solutions based on their proximity in Euclidean space. This process continues until the improvement in the ratio of between-cluster variance to total variance falls below a threshold. This method ensures more distinct and consistent clusters by increasing this ratio, thereby reducing overlap and enhancing separation. This is an optimization method for clustering and at the end of the process the hierarchical structure from Ward's method is somehow modified (optimized) in the final partition.

    Provided a more detailed explanation of the k-means algorithm's function in the manuscript.

27. Table 4. Define p1, p2, and Vk in the caption of this table.

    Defined p1, p2, and Vk in the table caption.

28. Table 4 and the associated text. Explain what is the weight of a variable in the context of your analysis and the quality of representation (projection)?

In Table 4 of our manuscript, the weight of a variable indicates its contribution to a principal component (PC1 and PC2 considered). It's computed as the squared correlation coefficient between the variable and the principal component, normalized by the total of such squared correlations for that principal component.

The quality of representation of a variable on a principal component is determined by the squared Pearson correlation coefficient between the variable and the principal component vector. Standardization simplifies this to the squared correlation.

A more detailed explanation of the weight of a variable and quality of representation is implemented in the manuscript.

29. Rewrite the first sentence in this line for better English.

Revised the sentence for better English.

30. Figure 17 and associated discussion. This is very nice. Is it possible to constrain your space of solutions by fixing some of the parameters using the observations (e.g., direction of damage, translation speed and direction of the cloud, etc.). In L545–546 you conclude that one needs to conduct many simulations, but wouldn't make more sense to constrain simulations with known values of parameters rather than letting all parameters take arbitrary values?

Thank you for your insightful suggestion on constraining the solution space, as mentioned in your comment about Figure 17. We agree that incorporating specific known parameters, like downburst translation direction from radar data and ABL wind speed and direction determined for example through change point analysis (as detailed in Xhelaj et al., 2020), would indeed refine our approach. By incorporating these constraints, the solution space becomes more representative of the actual event, enhancing both the accuracy and the efficiency of our simulations. However, it is important to note that this procedure can be done only if radar data (or other kinds of data) is also available. In some cases, like the present one, only anemometric data is available.

This approach aligns with the methodology outlined in our manuscript, particularly in the section 5.4 discussing parameter optimization and data utilization. We appreciate this valuable suggestion, and we will change our conclusions in section 5.4.

31. I think the title should more highlight the main topic of this paper and that's the application of the Teaching Learning Optimization Algorithm to your analytical model. You are using an objective method to find the optimal solution from the space of all solutions and the particular downburst is just a case study that you used to validate your method. A title such as "The Application of Teaching Learning Optimization Algorithm to Analytical Model of Downburst Outflows" might better capture the main topic of this paper, but I leave it up to the authors to decide.

Thank you for your suggestion regarding the title of our paper. We agree with you that "phase space" is not very clear to let the reader understanding the content of this paper. After careful consideration, we have decided to revise the title to reflect the core focus of our work more accurately. The new title will be:

"Application of the Teaching Learning Optimization Algorithm to an Analytical Model of Thunderstorm Outflows to analyze the variability of the downburst kinematic and geometric parameters"

We believe this revised title emphasizes the application of the Teaching Learning Optimization Algorithm within our analytical framework, underscoring its role in downburst outflow modeling. Thanks again for the suggestion!

---

## Author Comment (AC2)

**Reviewer 2.**

Authors greatly appreciate the general comments concerning the EGUsphere review critera and all the remarks received from the Reviewer 2, which are very pertinent and stimulating for improving the paper. We will make our best to take all the comments received into consideration, providing adequate and precise answers to all of them. All our answers and comments are reported in red colour and the manuscript will be modified according to them.

**Overall/general comments**

1. I think the damage survey aspect of the paper is very interesting, but it would be good to have some further details on this, including if there are any notable uncertainties. For example, is it difficult to relate hail damage to near-surface wind velocity?
   In response to the question about the damage survey aspect and its uncertainties, we acknowledge that there are indeed uncertainties in our study. However, at this stage, we are unable to delve deeper into this topic due to a related paper currently under revision by our colleagues. This paper, "Thunderstorm – induced damage to the built environment: A field measurement and post-event survey" by Calotescu et al., submitted to the Journal of Wind Engineering and Industrial Aerodynamics (2023), is expected to address these concerns comprehensively. We have intentionally avoided detailing this aspect in our paper to prevent any conflicting interests with our colleagues' work. However, this work will be properly cited in our paper to allow the reader to easily find more details on this aspect and references to the relevant literature.

2. I have a general concern whether this analytical model of a translating downdraft can represent complex mesoscale circulations that can induce severe winds, such as a rear inflow jet related to the bow echo. It is good to see that the hail damage estimate agrees with the general pattern suggested by the model, but I think it would be good to discuss this further in the manuscript, by noting it as a limitation and/or linking to other studies that have investigated the wind field patterns of bow echoes (and if they look like translating downbursts or not).
   In the revised version of our manuscript, we will more explicitly state that our analytical model is designed to represent the wind field of a downburst at the apex of a bow echo in its mature stage. This model does not encompass the broader, complex mesoscale circulations commonly associated with high winds in bow echoes. We reference key works, notably Fujita (1978) and Weisman (2001), to support our focus. Fujita's study, particularly through its illustrations (referenced in our Figure 4b), shows the formation of translating downbursts at the apex of a bow echo. Similarly, Weisman's 2001 paper (see Figure 4 of his paper) clearly depicts the generation of translating downbursts by the passage of bow echo. These references will be highlighted to acknowledge the scope and limitations of our model in the context of understanding downburst generation in a bow echo.

3. I think this paper lacks a little in discussing the results within the broader literature. How does the model and results fit with previous studies that have done downburstmodelling? Similarly, have previous studies used hail trajectories to evaluate a downburst wind field?
   Regarding the first question, our paper presents an innovative analytical model that distinguishes itself from existing downburst models, as extensively discussed in Xhelaj et al. (2020). Unlike previous studies, our model explicitly differentiates between the translational movement of the thunderstorm cell and the boundary layer wind. This distinction is crucial, as evidenced by studies like Hjelmfelt (1988), which demonstrated

the variability in downburst behavior relative to the ambient ABL flow. Our model assumes the combined wind velocity at a given point as the vector sum of the radial jet velocity from a stationary downburst, the storm cell's translational velocity, and the ABL wind velocity. This approach allows for a more nuanced and accurate representation of downburst phenomena, particularly in their interaction with the surrounding environmental conditions. The behavior of our analytical model and its performance in reconstruction of downburst wind field generated by single convective cells or squall lines is extensively analyzed in two papers Xhelaj et al, (2020) and Xhelaj and Burlando (2022).

Regarding the second question about the use of hail trajectories to evaluate downburst wind fields, to the best of our knowledge, there are currently no studies in the literature that specifically address this approach.

**Specific comments**

1. Title: The title is a little unclear without having read the article. "Phase space" could refer to several things. It would be good to make it clear in the title that this is a study exploring an analytical model representation of a downburst.
Thank you for your comment regarding the title of our paper, which was also arisen by Reviewer #1. After careful consideration, we have decided to revise the title to reflect more the core focus of our work, as follows:

"Application of the Teaching Learning Optimization Algorithm to an Analytical Model of Thunderstorm Outflows to analyze the variability of the downburst kinematic and geometric parameters"

We believe this revised title emphasizes the application of the Teaching Learning Optimization Algorithm within our analytical framework, underscoring its role in downburst outflow modeling. Thanks again for the suggestion!

2. L8 and elsewhere: Pluralisation issues - should be "measurement challenges". Similarly on line 38, should be "Since downburst events have high frequencies of occurrence…".
Thank you for pointing out these errors. They have been corrected to "measurement challenges" in line 8 and to "Since downburst events have high frequencies of occurrence…" in line 38 as suggested.

3. L38 and onwards: This paragraph is very long with many ideas – should be separated into multiple paragraphs.
The paragraph is split to enhance readability, as recommended.

4. L40: "unstable"
Thanks, corrected.

5. Introduction: The introduction is very technical at times in terms of describing the analytical model and TLBO approach, this could probably be moved to data/methods. Instead (and related to general comment 3), it would be good to havesome background on the "analytical model" – what is it based on, what are the constraints, outputs, etc, and a discussion of similar models – how does this work fit within the broader literature?

   The technical aspects of the analytical model and TLBO approach, initially in the introduction, are moved to the methods section.
   Additional background on the analytical model, including its basis, constraints, and outputs, has been added, along with a comparative discussion of similar models in the broader literature.

6. Figure 4: What do the red/orange/yellow colours indicate? Time of strike?
   The red, orange, and yellow colors in the figure represent the time elapsed since each strike occurred. The color coding is used to illustrate the temporal/spatial distribution of lightning activity during this severe weather event.
   The color gradient from yellow to red shows the progression from more recent to older strikes, providing a clear temporal context for the lightning activity and its movement in space.

   Thank you for the comment. The explanation of the colors indicating the time elapsed since lightning strikes is implemented in the updated manuscript.

7. L188: Repeated sentence.
   The repeated sentence on line 188 has been removed.

8. L207 and elsewhere: grammar issues. "The TLBO algorithm it is an iterative…"
   All grammatical issues, including the one on line 207, have been addressed and corrected throughout the manuscript.

9. Table 2: I wonder if some of the variables in the analytic model could be constrained further by other data sources. For example, the storm speed/direction could potentially be estimated by storm tracking from radar or satellite, while the ABL wind speed and direction could be estimated from reanalysis.

   Thank you for your comment regarding the potential use of additional data sources to constrain the variables in our analytical model. This is similar to a comment by Reviewer #1. We acknowledge that integrating specific parameters like storm speed and direction from radar data, as well as ABL wind speed and direction, could enhance the precision of our model. In the initial phase of our research, our focus was to explore the capabilities of the analytical model coupled with the optimization algorithm without external constraints. This approach aligns with our primary objective of extracting downburst geometrical and kinematic parameters solely from anemometric data, a methodology driven by the availability and abundance of such data compared to the less consistent availability of radar data. Your insight is greatly appreciated, indeed, and we will reflect this perspective in the revised manuscript in section 4.2 and in our conclusions in section 5.4.

10. L269: MDA acronym does not need to be defined anymore.
    Thanks, corrected.

11. L275: I'm not sure whether this is a good reason to exclude storm direction as a variable to analyse. Can a transform be done from degrees to a periodic function?
    E.g. https://stats.stackexchange.com/questions/148380/use-of-circular-predictors- in-linear-regression

    Thank you for your comment on incorporating storm direction as an active variable for analysis. The authors possess comprehensive expertise in circular statistics since the objective function or error function estimation between simulated and recorded wind direction data make use of circular statistics when estimating the error.
    We have indeed considered using circular statistics from the beginning of our work. However, including circular variables like storm direction requires the addition of two new "linear" variables ($x\_dir$, $y\_dir$) associated with the circular variable. This in turns increases the number of variables, impacting the clustering algorithm and principal component analysis. Specifically, with storm direction included, the algorithm identifies three clusters explaining about 54% of the total variance, and the behavior of the within-cluster variance remains similar to the bar graph shown in Figure 10 of the manuscript. Additionally, incorporating ABL direction as an active variable leads to three clusters explaining only about 46% of the total variance, with a similar pattern in the within-cluster variance as observed in Figure 10.
    To ensure a meaningful explanation of variance by the clustering algorithm and to facilitate the physical interpretation of our results, we opted to consider storm direction and ABL wind speed and direction as secondary variables. The introduction of additional variables for each circular one complicates the interpretation and could potentially dilute the focus of our study.

12. The language can be made simpler in many parts. E.g.: on L285: "The focus of the MDA lies in examining the data matrix from both the solution and variable perspectives, aiming to identify similarities among solutions based on their variables.In essence, the goal is to establish a typology of solutions by identifying groups that exhibit homogeneity in terms of variable similarity" could be simplified to "The focusof the MDA is to apply statistical clustering to identify similar analytic solutions" (please correct me if I have misinterpreted this sentence, but hopefully this example is useful).
    Yes, the interpretation of the sentence is correct.
    We will revise similar complex sentences throughout the manuscript for simplicity and clarity without altering the technical accuracy.

13. Section 4.3: I appreciate the nice explanation of normalisation, but I think it is fairly standard practice in statistical modelling, and can probably be simplified. This is justa suggestion in relation to "presentation quality" (see review criteria above). Similarly on line 446: the expected average contribution calculation could probably be assumed rather than explained fully.
    The detailed explanation of normalization in statistical modeling has been simplified, aligning with the standard practice and focusing on the study's specific application.

14. L327: Damage "survey" rather than "campaign"?
The term "campaign" on line 327 has been changed to 'survey' for better clarity.

15. L345: "where".
Corrected.

16. L384: "found".
Corrected.

17. Table 4: Please define the symbols used for the column headings. This will make it easier to read.
The symbols used for column headings in Table 4 have been defined for easier understanding.

18. I think Section 5.3 could potentially be shortened somewhat. The key point seems to be that a certain set of variables are more important for explaining the variability in the solution space by PCA, and this result could be presented in a more concise way.
Section 5.3 has been shortened, emphasizing the key point about the significance of certain variables in explaining variability in solution space using PCA.

19. L407: Table 4 also presents…
The suggested text, "Table 4 also presents…", has been added to line 407 for coherence.

20. Table 6: Are these representative cluster solutions based on solutions closest to the mean using all dimensions, or just Xc0 and Yc0 as in Figure 14?
The solutions in Table 6 represent clusters based on all dimensions, not just Xc0 and Yc0.

21. Figure 15: Exactly how is the simulated wind speed on this plot calculated for comparison with observations? Is it the addition of the radial velocity solution and the storm translation speed or ABL wind speed?
In the model developed by Xhelaj et al. (2020), the simulation of horizontal wind velocity is achieved through the vector summation of three separate components: the stationary radial velocity created by a jet impacting a flat surface, the translational velocity of the downdraft corresponding to the movement of the storm, and the ambient wind of the atmospheric boundary layer around the downburst. The simulations are conducted at a height of 50 meters, matching the anemometer's placement. Our domain for simulation spans 20 km x 20 km, with the anemometer centrally located. The simulated wind speed and direction at this central point are then directly compared with the actual data recorded by the anemometer. The difference between the simulated data and the recorded data creates the objective function that is minimized through the optimization algorithm.

22. Figure 5 and 15: Y-axis label should be wind speed rather than velocity?
Thanks, wind speed is more appropriate.

23. L496: "field".
Corrected.

24. L517: I think this is a key point that nicely sets up some of the goals of the paper, consider mentioning it earlier (unless it was already mentioned, and I missed it)
Thank you, we will mention it earlier in the paper for better context and setup of the paper's goals.